# ARTIFICIAL KURAMOTO OSCILLATORY NEURONS

**Takeru Miyato[1], Sindy Löwe[2], Andreas Geiger[1], Max Welling[2]**
[1] University of Tübingen, Tübingen AI Center   [2] University of Amsterdam

## ABSTRACT

It has long been known in both neuroscience and AI that "binding" between neurons leads to a form of competitive learning where representations are compressed in order to represent more abstract concepts in deeper layers of the network. More recently, it was also hypothesized that dynamic (spatiotemporal) representations play an important role in both neuroscience and AI. Building on these ideas, we introduce Artificial Kuramoto Oscillatory Neurons (*AKOrN*) as a dynamical alternative to threshold units, which can be combined with arbitrary connectivity designs such as fully connected, convolutional, or attentive mechanisms. Our generalized Kuramoto updates bind neurons together through their synchronization dynamics. We show that this idea provides performance improvements across a wide spectrum of tasks such as unsupervised object discovery, adversarial robustness, calibrated uncertainty quantification, and reasoning. We believe that these empirical results show the importance of rethinking our assumptions at the most basic neuronal level of neural representation, and in particular show the importance of dynamical representations. Code: https://github.com/autonomousvision/akorn. Project page: https://takerum.github.io/akorn_project_page/.

## 1 INTRODUCTION

Before the advent of modern deep learning architectures, artificial neural networks were inspired by biological neurons. In contrast to the McCulloch-Pitts neuron (McCulloch & Pitts, 1943) which was designed as an abstraction of an integrate-and-fire neuron (Sherrington, 1906), recent building blocks of neural networks are designed to work well on modern hardware (Hooker, 2021). As our understanding of the brain is improving over recent years, and neuroscientists are discovering more about its information processing principles, we can ask ourselves again if there are lessons from neuroscience that can be used as design principles for artificial neural nets.

In this paper, we follow a more modern dynamical view of neurons as oscillatory units that are coupled to other neurons (Muller et al., 2018). Similar to how the binary state of a McCulloch-Pitts neuron abstracts the firing of a real neuron, we will abstract an oscillating neuron by an $N$-dimensional unit vector that rotates on the sphere (Löwe et al., 2023). We build a new neural network architecture that has iterative modules that update $N$-dimensional oscillatory neurons via a generalization of the well-known non-linear dynamical model called the Kuramoto model (Kuramoto, 1984).

The Kuramoto model describes the synchronization of oscillators; each Kuramoto update applies forces to connected oscillators, encouraging them to become aligned or anti-aligned. This process is similar to binding in neuroscience and can be understood as distributed and continuous clustering. Thus, networks with this mechanism tend to compress their representations via synchronization.

We incorporate the Kuramoto model into an artificial neural network, by applying the differential equation that describes the Kuramoto model to each individual neuron. The resulting artificial Kuramoto oscillatory neurons (*AKOrN*) can be combined with layer architectures such as fully connected layers, convolutions, and attention mechanisms.

We explore the capabilities of *AKOrN* and find that its neuronal mechanism drastically changes the behavior of the network. *AKOrN* strongly binds object features with competitive performance to slot-based models in object discovery, enhances the reasoning capability of self-attention, and increases robustness against random, adversarial, and natural perturbations with surprisingly good calibration.

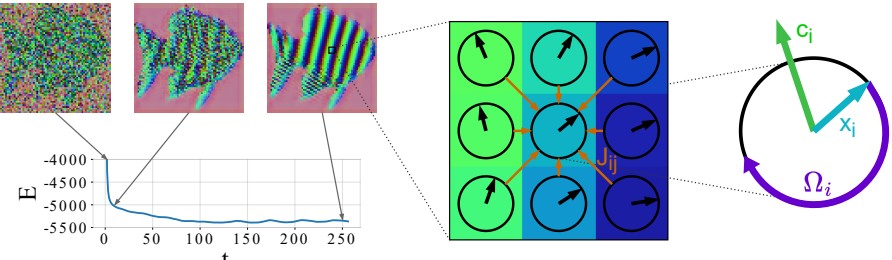

Figure 1: Our proposed artificial Kuramoto oscillatory neurons (*AKOrN*). The series of pictures on the left are $64 \times 64$ oscillators evolving by the Kuramoto updates (Eq. (2)), along with a plot of the energies computed by Eq. (3). Each single oscillator $\mathbf{x}_i$ is an $N$-dimensional vector on the sphere and is influenced by (1) connected oscillators through the weights $\mathbf{J}_{ij}$, (2) conditional stimuli $\mathbf{c}_i$, and (3) $\mathbf{\Omega}_i$ that determines the natural frequency of each oscillator. See Fig. 10 for details on $\mathbf{C}$ and $\mathbf{J}$.

## 2 MOTIVATION

It was recognized early on that neurons interact via lateral connections (Hubel & Wiesel, 1962; Somers et al., 1995). In fact, neighboring neurons tend to cluster their activities (Gray et al., 1989; Mountcastle, 1997), and clusters tend to compete to explain the input. This "competitive learning" has the advantage that information is compressed as we move through the layers, facilitating the process of abstraction by creating an information bottleneck (Amari & Arbib, 1977). Additionally, the competition encourages different higher-level neurons to focus on different aspects of the input (i.e. they specialize). This process is made possible by synchronization: like fireflies in the night, neurons tend to synchronize their activities with their neighbors', which leads to the compression of their representations. This idea has been used in artificial neural networks before to model "binding" between neurons, where neurons representing features such as square, blue, and toy are bound by synchronization to represent a square blue toy (Mozer et al., 1991; Reichert & Serre, 2013; Löwe et al., 2022). In this paper, we will use an $N$-dimensional generalization of the famous Kuramoto model (Kuramoto, 1984) to model this synchronization.

Our model has the advantage that it naturally incorporates spatiotemporal representations in the form of traveling waves (Keller et al., 2024), for which there is ample evidence in the neuroscientific literature. While their role in the brain remains poorly understood, it has been postulated that they are involved in short-term memory, long-range coordination between brain regions, and other cognitive functions (Rubino et al., 2006; Lubenov & Siapas, 2009; Fell & Axmacher, 2011; Zhang et al., 2018; Roberts et al., 2019; Muller et al., 2016; Davis et al., 2020; Benigno et al., 2023). For example, Muller et al. (2016) finds that oscillatory patterns in the thalamocortical network during sleep are organized into circular wave-like patterns, which could give an account of how memories are consolidated in the brain. Davis et al. (2020) suggest that spontaneous traveling waves in the visual cortex modulate synaptic activities and thus act as a gating mechanism in the brain. In the generalized Kuramoto model, traveling waves naturally emerge as neighboring oscillators start to synchronize (see on the left in Fig. 1, and Fig. 10 in the Appendix).

Another advantage of using dynamical neurons is that they can perform a form of reasoning. Kuramoto oscillators have been successfully used to solve combinatorial optimization tasks such as k-SAT problems (Heisenberg, 1985; Wang & Roychowdhury, 2017). This can be understood by the fact that Kuramoto models can be viewed as continuous versions of discrete Ising models, where phase variables replace the discrete spin states. Many authors have argued that the modern architectures based on, e.g., transformers lack this intrinsic capability of "neuro-symbolic reasoning" (Dziri et al., 2024; Bounsi et al., 2024). We show that *AKOrN* can successfully solve Sudoku puzzles, illustrating this capability. Additionally, *AKOrN* relates to models in quantum physics and active matter (see appendix B.1).

In summary, *AKOrN* combines beneficial features such as competitive learning (i.e., feature binding), reasoning, robustness and uncertainty quantification, as well as the potential advantages of traveling waves observed in the brain, while being firmly grounded in well-understood physics models.

## 3 THE KURAMOTO MODEL

The Kuramoto model (Kuramoto, 1984) is a non-linear dynamical model of oscillators, that exhibits synchronization phenomena. Even with its simple formulation, the model can represent numerous dynamical patterns depending on the connections between oscillators (Breakspear et al., 2010; Heitmann et al., 2012).

In the original Kuramoto model, each oscillator $i$ is represented by its phase information $\theta_i \in [0, 2\pi)$. The differential equation of the Kuramoto model is

$$\dot{\theta}_i = \omega_i + \sum_j J_{ij} \sin(\theta_j - \theta_i), \tag{1}$$

where $\omega_i \in \mathbb{R}$ is the natural frequency and $J_{ij} \in \mathbb{R}$ represents the connections between oscillators: if $J_{ij} > 0$ the $i$ and $j$-th oscillator tend to align, and if $J_{ij} < 0$, they tend to oppose each other.

While the original Kuramoto model describes one-dimensional oscillators, we use a *multi-dimensional vector version* of the model (Olfati-Saber, 2006; Zhu, 2013; Chandra et al., 2019; Lipton et al., 2021; Markdahl et al., 2021) with a symmetry-breaking term into neural networks. We denote oscillators by $\mathbf{X} = \{\mathbf{x}_i\}_{i=1}^C$, where each $\mathbf{x}_i$ is a vector on a hypersphere: $\mathbf{x}_i \in \mathbb{R}^N$, $\|\mathbf{x}_i\|_2 = 1$. $N$ is each single oscillator dimension called *rotating dimensions* and $C$ is the number of oscillators. While each $\mathbf{x}_i$ is time-dependent, we omit $t$ for clarity. The oscillator index $i$ may have multiple dimensions: if the input is an image, for example, each oscillator is represented by $\mathbf{x}_{c,h,w}$ with $c, h, w$ indicating channel, height and width positions, respectively.

The differential equation of our vector-valued Kuramoto model is written as follows:

$$\dot{\mathbf{x}}_i = \mathbf{\Omega}_i \mathbf{x}_i + \mathrm{Proj}_{\mathbf{x}_i}(\mathbf{c}_i + \sum_j \mathbf{J}_{ij} \mathbf{x}_j) \text{ where } \mathrm{Proj}_{\mathbf{x}_i}(\mathbf{y}_i) = \mathbf{y}_i - \langle \mathbf{y}_i, \mathbf{x}_i \rangle \mathbf{x}_i \tag{2}$$

Here, $\mathbf{\Omega}_i$ is an $N \times N$ anti-symmetric matrix and $\mathbf{\Omega}_i \mathbf{x}_i$ is called the natural frequency term that determines each oscillator's own rotation frequency and angle. The second term governs interactions between oscillators, where $\mathrm{Proj}_{\mathbf{x}_i}$ is an operator that projects an input vector onto the tangent space of the sphere at $\mathbf{x}_i$. We show a visual description of $\mathrm{Proj}_{\mathbf{x}_i}$ and a relation between the vector valued Kuramoto model and the original one in the Appendix A.1. $\mathbf{C} = \{\mathbf{c}_i\}_{i=1}^C, \mathbf{c}_i \in \mathbb{R}^N$ is a data-dependent variable, which is computed from the observational input or the activations of the previous layer. In this paper, every $\mathbf{c}_i$ is set to be constant across time, but it can be a time-dependent variable. $\mathbf{c}_i$ can be seen as another oscillator that has a unidirectional connection to $\mathbf{x}_i$. Since $\mathbf{c}_i$ is not affected by any oscillators, $\mathbf{c}_i$ strongly binds $\mathbf{x}_i$ to the same direction as $\mathbf{c}_i$, i.e. it acts as a bias direction (see Fig. 10 in the Appendix). In physics lingo, $\mathbf{C}$ is often referred to as a "symmetry breaking" field.

The Kuramoto model is Lyapunov if we assume certain symmetric properties in $\mathbf{J}_{ij}$ and $\mathbf{\Omega}_i$ (Aoyagi, 1995; Wang & Roychowdhury, 2017). For example, if $\mathbf{J}_{ij} = J_{ij}\mathbf{I}$, $J_{ij} = J_{ji} \in \mathbb{R}$, $\mathbf{\Omega}_i = \mathbf{\Omega}$, and $\mathbf{\Omega}\mathbf{c}_i = \mathbf{0}$, each update is guaranteed to minimize the following energy (proof is found in Sec F):

$$E = -\frac{1}{2} \sum_{i,j} \mathbf{x}_i^{\mathrm{T}} \mathbf{J}_{ij} \mathbf{x}_j - \sum_i \mathbf{c}_i^{\mathrm{T}} \mathbf{x}_i \tag{3}$$

Fig. 1 on the left shows how the oscillators and the corresponding energy evolve with a simple Gaussian kernel as the connectivity matrix. Here, we set $\mathbf{C}$ as a silhouette of a fish, where $\mathbf{c}_i = \mathbf{1}$ on the outer silhouette and $\mathbf{c}_i = \mathbf{0}$ on the inner silhouette. The oscillator state is initially disordered, but gradually exhibits collective behavior, eventually becoming a spatially propagating wavy pattern. We include animations of visualized oscillators, including oscillators of trained *AKOrN* models used in our experiments, in the Supplementary Material.

We would like to note that we found that even without symmetric constraints, the energy value decreases relatively stably, and the models perform better across all tasks we tested compared to models with symmetric $\mathbf{J}$. A similar observation is made by Effenberger et al. (2022) where heterogeneous oscillators such as those with different natural frequencies are helpful for the network to control the level of synchronization and increase the network capacity. From here, we assume no symmetric constraints on $\mathbf{J}$ and $\mathbf{\Omega}$. Having asymmetric (a.k.a. non-reciprocal) connections is aligned with the biological neurons in the brain, which also do not have symmetric synapses.

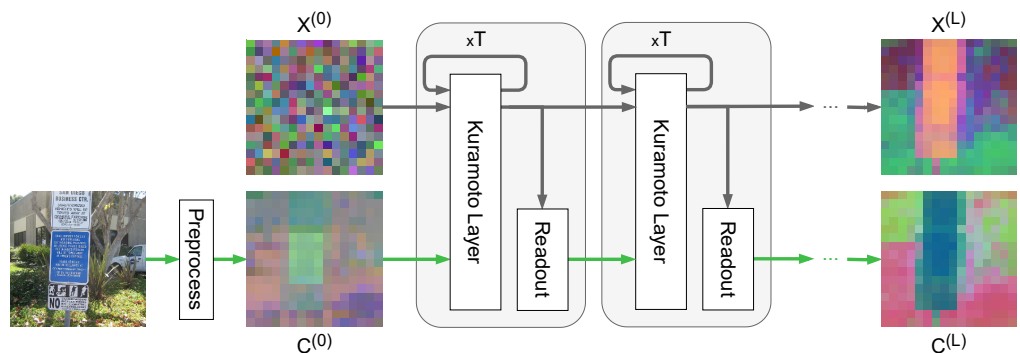

Figure 2: Our proposed Kuramoto-based network (here, for image processing). Each block consists of a Kuramoto-layer and a readout module described in Sec 4. $\mathbf{C}^{(L)}$ is used to make the final prediction of our model. Similar network structures are proposed in (Bansal et al., 2022; Geiping et al., 2025).

## 4 NETWORKS WITH KURAMOTO OSCILLATORS

We utilize the artificial Kuramoto oscillator neurons (*AKOrN*) as a basic unit of information processing in neural networks (Fig. 2). First, we transform an observation with a relatively simple function to create the initial conditional stimuli $\mathbf{C}^{(0)}$. Next, $\mathbf{X}^{(0)}$ is initialized by either $\mathbf{C}^{(0)}$, a fixed learned embedding, random vectors, or a mixture of these initialization schemes. The block is composed of two modules: the Kuramoto layer and the readout module, which together process the pair $\{\mathbf{X}, \mathbf{C}\}$. The Kuramoto layer updates $\mathbf{X}$ with the conditional stimuli $\mathbf{C}$, and the readout layer extracts features from the final oscillatory states to create new conditional stimuli. We denote the number of layers by $L$, and $l$-th layer's output by $\{\mathbf{X}^{(l)}, \mathbf{C}^{(l)}\}$.

**Kuramoto layer** Starting with $\mathbf{X}^{(l,0)} := \mathbf{X}^{(l-1)}$ as initial oscillators, where the second superscript denotes the time step, we update them by the discrete version of the differential equation (2):

$$\Delta\mathbf{x}_i^{(l,t)} = \mathbf{\Omega}_i^{(l)}\mathbf{x}_i^{(l,t)} + \text{Proj}_{\mathbf{x}_i^{(l,t)}}(\mathbf{c}_i^{(l-1)} + \sum_j \mathbf{J}_{ij}^{(l)}\mathbf{x}_j^{(l,t)}) \tag{4}$$

$$\mathbf{x}_i^{(l,t+1)} = \Pi\left[\mathbf{x}_i^{(l,t)} + \gamma\Delta\mathbf{x}_i^{(l,t)}\right], \tag{5}$$

where $\Pi$ is the normalizing operator $\mathbf{x}/\|\mathbf{x}\|_2$ that ensures that the oscillators stay on the sphere. $\gamma > 0$ is a scalar controlling the step size of the update, which is learned in our experiments. We call this update a Kuramoto update or a Kuramoto step from here. We optimize both $\mathbf{\Omega}^{(l)}$ and $\mathbf{J}^{(l)}$ given the task objective.

We update the oscillators $T$ times. We denote the oscillators at $T$ by $\mathbf{X}^{(l,T)}$. This oscillator state is used as the initial state of the next block: $\mathbf{X}^{(l)} := \mathbf{X}^{(l,T)}$.

**Readout module** We read out patterns encoded in the oscillators to create new conditional stimuli $\mathbf{C}^{(l)}$ for the subsequent block. Since the oscillators are constrained onto the (unit) hyper-sphere, all the information is encoded in their directions. In particular, the relative direction between oscillators is an important source of information because patterns after certain Kuramoto steps only differ in global phase shifts (see the last two patterns in Fig. 10 in the Appendix). To capture phase invariant patterns, we take the norm of the linearly processed oscillators:

$$\mathbf{C}^{(l)} = g(\mathbf{m}) \in \mathbb{R}^{C' \times N}, m_k = \|\mathbf{z}_k\|_2, \ \mathbf{z}_k = \sum_i \mathbf{U}_{ki}\mathbf{x}_i^{(l,T)} \in \mathbb{R}^{N'}, \tag{6}$$

where $\mathbf{U}_{ki} \in \mathbb{R}^{N' \times N}$ is a learned weight matrix, $g$ is a learned function, and $\mathbf{m} = [m_1, ..., m_K]^{\text{T}} \in \mathbb{R}^K$. $N'$ is typically set to the same value as $N$. In this work, $g$ is just the identity function, a linear layer, or at most a three-layer neural network with residual connections. Because the module computes the norm of (weighted) $\mathbf{X}^{(l,T)}$, this readout module includes functions that are invariant to the global phase shift in the solution space. Unless otherwise specified, we set $C' = C$ and $K = C \times N$ in all our experiments.

### 4.1 CONNECTIVITIES

We implement artificial Kuramoto oscillator neurons (*AKOrN*) within convolutional and self-attention layers. We write down the formal equations of the connectivity for completeness, however, they simply follow the conventional operation of convolution or self-attention applied to oscillatory neurons flattened w.r.t the rotating dimension $N$. In short, convolutional connectivity is local, and attentive connectivity is dynamic input-dependent connectivity.

**Convolutional connectivity** To implement *AKOrN* in a convolutional layer, oscillators and conditional stimuli are represented as $\{\mathbf{x}_{c,h,w}, \mathbf{c}_{c,h,w}\}$ where $c, h, w$ are channel, height and width positions, and the update direction is given by:

$$\mathbf{y}_{c,h,w} := \mathbf{c}_{c,h,w} + \sum_d \sum_{h',w' \in R[H',W']} \mathbf{J}_{c,d,h',w'} \mathbf{x}_{d,(h+h'),(w+w')}, \tag{7}$$

where $R[H', W'] = [1, ..., H'] \times [1, ..., W']$ is the $H' \times W'$ rectangle region (i.e. kernel size) and $\mathbf{J}_{c,d,h',w'} \in \mathbb{R}^{N \times N}$ are the learned weights in the convolution kernel where $(c, d), (h', w')$ are output and input channels, and height and width positions.

**Attentive connectivity** Similar to Bahdanau et al. (2014); Vaswani et al. (2017), we construct the internal connectivity in the QKV-attention manner. In this case, oscillators and conditional stimuli are represented by $\{\mathbf{x}_{l,i}, \mathbf{c}_{l,i}\}$ where $l$ and $i$ are indices of tokens and channels, respectively. The update direction becomes:

$$\mathbf{y}_{l,i} := \mathbf{c}_{l,i} + \sum_{m,j} \mathbf{J}_{l,m,i,j} \mathbf{x}_{m,j} = \mathbf{c}_{l,i} + \sum_{m,j} \sum_{k,h} \mathbf{W}^O_{h,i,k} A_h(l,m) \mathbf{W}^V_{h,k,j} \mathbf{x}_{m,j} \tag{8}$$

$$A_h(l,m) = \frac{e^{d_h(l,m)}}{\sum_m e^{d_h(l,m)}}, \ d_h(l,m) = \sum_a \left\langle \sum_i \mathbf{W}^Q_{h,a,i} \mathbf{x}_{l,i}, \sum_i \mathbf{W}^K_{h,a,i} \mathbf{x}_{m,i} \right\rangle \tag{9}$$

where $\mathbf{W}^O_{h,i,k}, \mathbf{W}^V_{h,k,j}, \mathbf{W}^Q_{h,a,i}, \mathbf{W}^K_{h,a,i} \in \mathbb{R}^{N \times N}$ are learned weights of head $h$. Since the connectivity is dependent on the oscillator values and thus not static during the updates, it is unclear whether the energy defined in Eq, (3) is proper. Nonetheless, in our experiments, the energy and oscillator states are stable after several updates (see the Supplementary Material, which includes visualizations of the oscillators of trained *AKOrN* models and their corresponding energies over timesteps).

## 5 RELATED WORKS

Many studies have historically incorporated oscillatory properties into artificial neural networks (Baldi & Pineda, 1991; Wang & Terman, 1997; Ketz et al., 2013; Neil et al., 2016; Chen et al., 2021b; Rusch & Mishra, 2020; Laborieux & Zenke, 2022; Rusch et al., 2022; van Gerven & Jensen, 2024). The Kuramoto model, a well-known oscillator model describing synchronization phenomena, has been rarely explored in machine learning, particularly in deep learning. However, several works motivate us to use the Kuramoto model as a mechanism for learning binding features. For example, although tested only in fairly synthetic settings, Liboni et al. (2023) show that cluster features emerge in the oscillators of the Kuramoto model with lateral connections without optimization. Ricci et al. (2021) studies how data-dependent connectivity can construct synchrony on synthetic examples. Nguyen et al. (2024) relates the over-smoothing to the notion of phase-synchrony and uses the model to mitigate over-smoothing phenomena in graph neural networks. Also, a line of works on neural synchrony (Reichert & Serre, 2013; Löwe et al., 2022; Stanić et al., 2023; Zheng et al., 2023; Löwe et al., 2023; Gopalakrishnan et al., 2024) shares the same philosophy with *AKOrN*. Zheng et al. (2023) model synchrony by using temporal spiking neurons based on biological neuronal mechanisms. Löwe et al. (2023) extend the concept of complex-valued neurons—used by Reichert & Serre (2013); Löwe et al. (2022) to abstract temporal neurons—into multidimensional neurons. They show that, together with a specific activation function called *χ-binding* that implements the 'winner-take-all' mechanism at the single neuron level (Löwe et al., 2024), the multidimensional neurons learn to encode binding information in their orientations. Those synchrony-based models are shown to work well on relatively synthetic data but have been struggling to scale to natural images. Löwe et al. (2023) show that their model can work with a large pre-trained self-supervised learning (SSL) model as a feature extractor, but its performance improvement is limited compared to slot-based models.

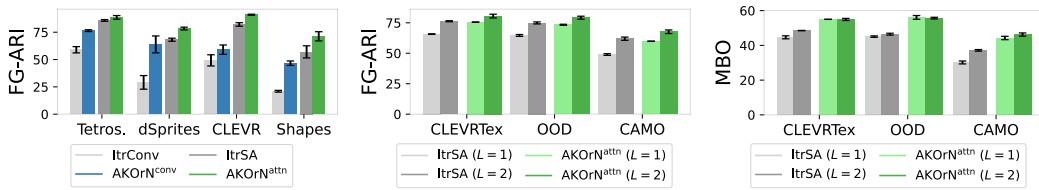

Figure 3: Object discovery performance on synthetic datasets.

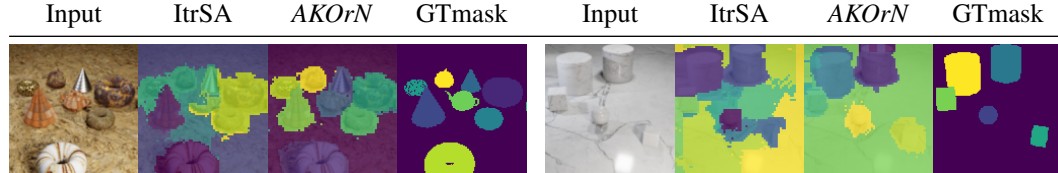

Figure 4: *AKOrN* learns more object-bound features than the non-Kuramoto model counterpart.

| Model | CLEVRTex | | OOD | | CAMO | |
|---|---|---|---|---|---|---|
| | FG-ARI | MBO | FG-ARI | MBO | FG-ARI | MBO |
| *MONet (Burgess et al., 2019) | $19.8_{\pm 1.0}$ | - | $37.3_{\pm 1.0}$ | - | $31.5_{\pm 0.9}$ | - |
| SLATE (Singh et al., 2022) | $44.2_{\pm NA}$ | $50.9_{\pm NA}$ | - | - | - | - |
| *Slot-Attetion (Locatello et al., 2020) | $62.4_{\pm 2.3}$ | - | $58.5_{\pm 1.9}$ | - | $57.5_{\pm 1.0}$ | - |
| Slot-diffusion (Wu et al., 2023) | $69.7_{\pm NA}$ | $61.9_{\pm NA}$ | - | - | - | - |
| Slot-diffusion+BO (Wu et al., 2023) | $78.5_{\pm NA}$ | $68.7_{\pm NA}$ | - | - | - | - |
| *DTI (Monnier et al., 2021) | $79.9_{\pm 1.4}$ | - | $73.7_{\pm 1.0}$ | - | $72.9_{\pm 1.9}$ | - |
| *I-SA (Chang et al., 2022) | $79.0_{\pm 3.9}$ | - | $83.7_{\pm 0.9}$ | - | $57.2_{\pm 13.3}$ | - |
| BO-SA (Jia et al., 2023) | $80.5_{\pm 2.5}$ | - | $86.5_{\pm 0.2}$ | - | $63.7_{\pm 6.1}$ | - |
| ISA-TS (Biza et al., 2023) | $92.9_{\pm 0.4}$ | - | $84.4_{\pm 0.8}$ | - | $86.2_{\pm 0.8}$ | - |
| *AKOrN*$^{\text{attn}}$ | $88.5_{\pm 0.9}$ | $59.7_{\pm 0.9}$ | $87.7_{\pm 0.3}$ | $60.8_{\pm 0.6}$ | $77.0_{\pm 0.5}$ | $53.4_{\pm 0.7}$ |

Table 1: Object discovery performance on CLEVRTex and its variants (OOD, CAMO). *AKOrN* is compared among models trained from scratch. *Numbers taken from Jia et al. (2023).

Slot-based models (Le Roux et al., 2011; Burgess et al., 2019; Greff et al., 2019; Locatello et al., 2020) are the most-used model for object-centric (OC) learning. Their discrete nature of representations is shown to be a good inductive bias to learn such OC representations. However, similarly to synchrony-based models, these models struggle on natural images and are therefore often combined with powerful, pre-trained SSL models such as DINO (Caron et al., 2021). Our proposed continuous Kuramoto neurons can be a building block of the SSL network itself, and we show that they learn better object-centric features than well-known SSL models. Our work is the first work that demonstrates that a synchrony-based model is solely scaled up to natural images.

*AKOrN*s perform particularly well on object discovery tasks when implemented in self-attention layers. Self-attention updates with normalization have been shown mathematically to cluster token features (Geshkovski et al., 2024). Our work combines this clustering behavior of transformers with the clustering induced by the synchronization of the Kuramoto neurons, resulting in *AKOrN* being the first competitive method to slot-based approaches.

Finally, there exist several works on interpreting self-attention in the context of the Hopfield networks (Ramsauer et al., 2020; Hoover et al., 2023). Energy transformer (Hoover et al., 2023) introduces a symmetrized attention mechanism to guarantee the update minimizing certain energy. However, we find such symmetric models worsen the performance in our reasoning task. Our Kuramoto-based models differ from these approaches: unit-norm-constrained neurons with asymmetric connections in $\mathbf{J}$, and their symmetry-breaking term $\mathbf{C}$. These elements contribute to performance improvement over the approach by (Hoover et al., 2023) and conventional self-attention in the reasoning task of our experiments.

| Input | DINO | *AKOrN* | GTMask | Input | DINO | *AKOrN* | GTmask |

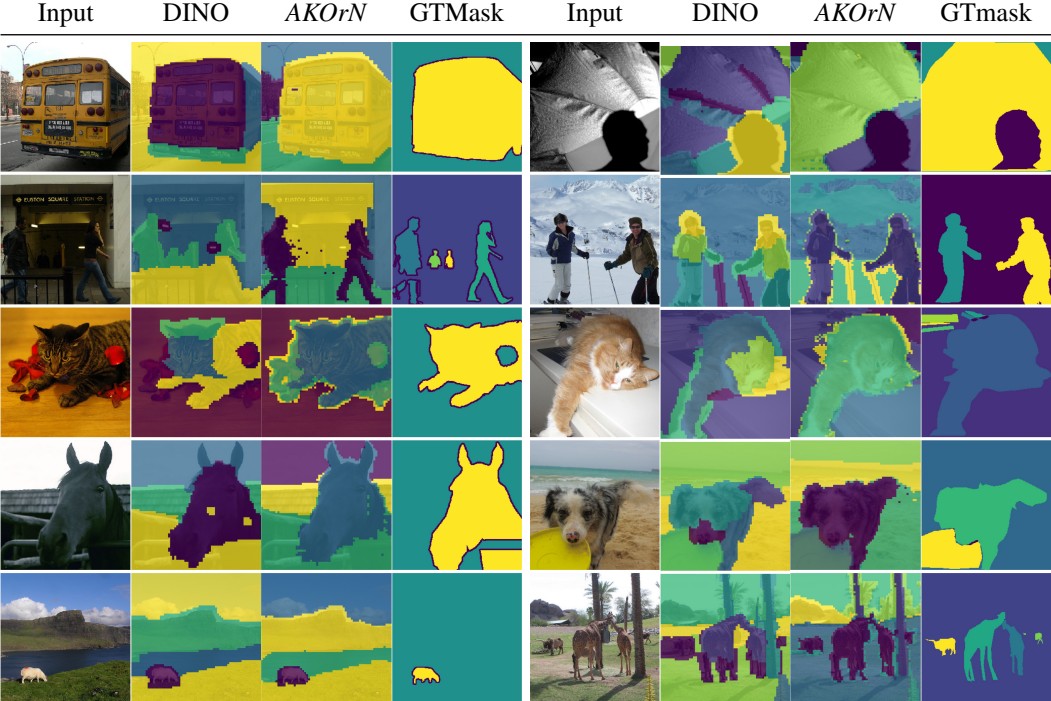

Figure 5: Visualization of clusters on (Left) PascalVOC and (Right) COCO2017.

# 6 EXPERIMENTS

## 6.1 UNSUPERVISED OBJECT DISCOVERY

Unsupervised object discovery is the task of finding objects in an image without supervision. Here, we test *AKOrN* on five synthetic datasets (Tetrominoes, dSprites, CLEVR (Kabra et al., 2019), Shapes, CLEVRTex (Karazija et al., 2021)) and two real image datasets (PascalVOC (Everingham et al., 2010), COCO2017 (Lin et al., 2014)) (see the Appendix C for details). Among the five synthetic datasets, CLEVRTex has the most complex objects and backgrounds. We further evaluate the models trained on the CLEVRTex dataset on two variants (OOD, CAMO). The materials and shapes of objects in OOD differ from those in CLEVRTex, while CAMO (short for camouflage) features scenes where objects and backgrounds share similar textures within each scene.

As baselines, we train models that are similar to ResNet (He et al., 2016) and ViT (Dosovitskiy et al., 2021), but iterate the convolution or self-attention layers multiple times with shared parameters. This allows us to evaluate the impact of our proposed, Kuramoto-based iterative updates. We denote these baselines as Iterative Convolution (ItrConv) and Iterative Self-Attention (ItrSA), respectively. Fig. 11 in the Appendix shows diagrams of each network. In *AKOrN*, $\mathbf{C}$ is initialized by the patched features of the images, while each $\mathbf{x}_i$ is initialized by random oscillators sampled from the uniform distribution on the sphere. We train the *AKOrN* model and baselines with the self-supervised SimCLR (Chen et al., 2020) objective.

We train each model from scratch on the five synthetic datasets. For the two real image datasets, we first train *AKOrN* on ImageNet (Krizhevsky et al., 2012) and directly evaluate that ImageNet-pretrained model on both datasets without fine-tuning. When evaluating, we apply clustering to the final block's output features (In *AKOrN*, it is $\mathbf{C}^{(L)}$). We use agglomeration clustering with average linkage, which we found to outperform K-means for both the baseline models and *AKOrN*. We evaluate the clustering results by foreground adjusted rand index (FG-ARI) and Mean-Best-Overlap (MBO). FG-ARI measures the similarity between the ground truth masks and the computed clusters, only for foreground objects. MBO first assigns each cluster to the highest overlapping ground truth mask and then computes the average intersection-over-union (IoU) of all pairs. See C.1.1 for details. For PascalVOC and COCO2017, we show instance-level MBO ($\text{MBO}_i$) and class-level ($\text{MBO}_c$) segmentation results.

| Model | PascalVOC | | COCO2017 | |
| | $MBO_i$ | $MBO_c$ | $MBO_i$ | $MBO_c$ |
| --- | --- | --- | --- | --- |
| (slot-based models) | | | | |
| Slot-attention | 22.2 | 23.7 | 24.6 | 24.9 |
| SLATE (Singh et al., 2021) | 35.9 | 41.5 | 29.1 | 33.6 |
| (DINO + slot-based model) | | | | |
| DINOSAUR (Seitzer et al., 2023) | 44.0 | 51.2 | 31.6 | 39.7 |
| Slot-diffusion (Wu et al., 2023) | 50.4 | 55.3 | 31.0 | 35.0 |
| SPOT (Kakogeorgiou et al., 2024) | 48.3 | 55.6 | **35.0** | **44.7** |
| (transformer + SSL) | | | | |
| MAE (He et al., 2022) | 34.0 | 38.3 | 23.1 | 28.5 |
| MoCoV3 (Chen et al., 2021a) | 47.3 | 53.0 | 28.7 | 36.0 |
| DINO (Caron et al., 2021) | 47.2 | 53.5 | 29.4 | 37.0 |
| *AKOrN* | **52.0** | **60.3** | 31.3 | 40.3 |

Table 2: Object discovery on PascalVOC and COCO2017.

Because the patched feature resolution is small due to patchification, the obtained cluster assignments are coarse. To compute finer cluster assignments, we introduce an upsampling method called *up-tiling*. This involves shifting the input image slightly along the horizontal and/or vertical axes to generate multiple feature maps. These feature maps are then interleaved to create a higher-resolution feature map. See Section C.1.2 in the Appendix for the methodological details of this up-tiling.

***AKOrN* binds object features** Fig. 3 shows that *AKOrN*s improve the object discovery performance over their non-Kuramoto counterparts in every dataset. Interestingly, we observe that convolution is less effective than attention. In Fig. 4, we see that the Kuramoto models' clusters are well-aligned with the individual objects, while clusters of the ItrSA model often span across objects and background, and are sensitive to the texture of the background and the specular highlight on the floor (more clustering results are shown in Figs 32-34 in the Appendix).

Tab. 1 shows a comparison to existing works on CLEVRTex and its variants. All other methods are slot-based. Among the distributed representation models, *AKOrN is the first method that is shown to be competitive with slot-based models on the complex CLEVRTex dataset*.

***AKOrN* scales to natural images** Fig. 5 shows *AKOrN* binds object features on natural images much better than DINO (Caron et al., 2021). We show a benchmark comparison on Pascal VOC and COCO2017 in Tab. 2. The proposed AKOrN model outperforms existing SSL models including DINO, MoCoV3, and MAE on both datasets, showing that it learns more object-bound features than conventional transformer-based models. On Pascal, *AKOrN* is considerably better than other models including models trained from scratch and models trained on features of a pretrained DINO model. On COCO, *AKOrN* again outperforms methods that are trained from scratch and is competitive to DINOSAUR and Slot-diffusion, but is outperformed by the recent SPOT model.

## 6.2 SOLVING SUDOKU

To test *AKOrN*'s reasoning capability, we apply it on the Sudoku puzzle datasets (Wang et al., 2019; Palm et al., 2018). The training set contains boards with 31-42 given digits. We test models in in-distribution (ID) and out-of-distribution (OOD) scenarios. The ID test set contains 1,000 boards sampled from the same distribution, while boards in the OOD set contain much fewer given digits (17-34) than the train set. To initialize $\mathbf{C}$, we use embeddings of the digits 0-9 (0 for blank, 1-9 for given digits). The initial $\mathbf{x}_i$ takes the value $\mathbf{c}_i/\|\mathbf{c}_i\|_2$ when a digit is given, and is randomly sampled from the uniform distribution on the sphere for blank squares. The number of Kuramoto steps during training is set to 16. We also train a transformer model with 8 blocks.

***AKOrN* solves Sudoku puzzles** *AKOrN* perfectly solves all puzzles in the ID test set, while only Recurrent Transformer (R-Transformer (Yang et al., 2023)) achieves this (Tab. 3). On the OOD set, *AKOrN* achieves 89.5±2.5 which is better than all other existing approaches including IRED (Du et al., 2024), an energy-based diffusion model. *AKOrN* again strongly outperforms its non-Kuramoto counterparts, ItrSA and Transformer.

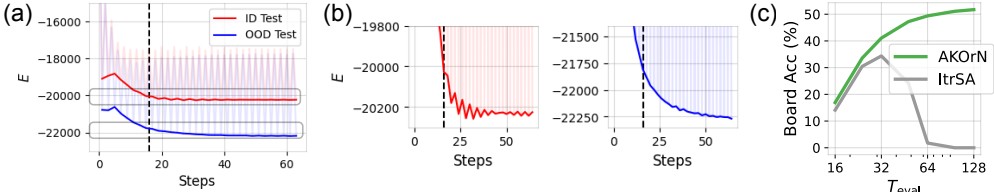

Figure 6: (a) Transition of the energy in Eq. (3) over # Kuramoto steps on the Sudoku datasets. The semi-transparent lines are actual energy values averaged across examples, and the solid ones connect the troughs. The dotted vertical line indicates # Kuramoto steps set during training. (b) A zoomed-in version of each plot. (c) The effect of test-time extension on # Kuramoto steps.

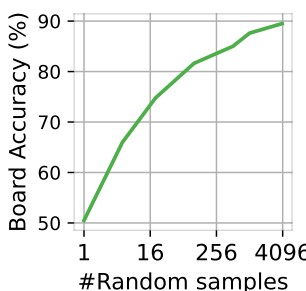

| Model | ID | OOD |
|---|---|---|
| SAT-Net (Wang et al., 2019) | 98.3 | 3.2 |
| Diffusion (Du et al., 2024) | 66.1 | 10.3 |
| IREM (Du et al., 2022) | 93.5 | 24.6 |
| RRN (Palm et al., 2018) | 99.8 | 28.6 |
| R-Transformer (Yang et al., 2023) | **100.0** | 30.3 |
| IRED (Du et al., 2024) | 99.4 | 62.1 |
| Transformer | $98.6_{\pm 0.3}$ | $5.2_{\pm 0.2}$ |
| ItrSA | $95.7_{\pm 8.5}$ | $34.4_{\pm 5.4}$ |
| *AKOrN*$^{\text{attn}}$ | $\mathbf{100.0_{\pm 0.0}}$ | $\mathbf{89.5_{\pm 2.5}}$ |

Figure 7: Improvement of board accuracy by the post-selection of predictions based on the $E$ values described in Sec 6.2. $T_{\text{eval}}$ is set to 128.

Table 3: Board accuracy on Sudoku Puzzles. We show the mean and std of the accuracy of models with 5 different random seeds for the weight initialization. The AKOrN results are obtained with $T_{\text{eval}} = 128$ and the energy-based voting with 4096 samples of initial oscillators.

**Test-time extension of the Kuramoto steps** Just as we humans use more time to solve harder problems, *AKOrN*'s performance improves as we increase the number of Kuramoto steps. As shown in Fig. 6 (a,b), on the ID test set, the energy fluctuates but roughly converges to a minimum after around 32 steps. On the OOD test set, however, the energy continues to decrease further. Fig. 6 (c) shows that increasing the number of Kuramoto steps at test time improves accuracy significantly (17% to 52%), while increasing the step count of standard self-attention provides a limited improvement on the OOD set (14% to 34%) and leads to lower performance on the ID set (99.3% to 95.7%).

**The energy value tells the correctness of the boards** The energy value defined in Eq (3) is a good indicator of the solution's correctness. In fact, we observe that predictions with low-energy oscillator states tend to be correct (see Fig. 28). We utilize this property to improve the performance. For each given board, we sample multiple predictions with different initial oscillators and select the lowest-energy prediction as the model's answer, which we call *Energy-based voting* ($E$-vote). We see in Fig. 7 that by increasing the number of sampled predictions, the model's board accuracy improves. This result implies that the Kuramoto layer behaves like *energy-based models*, even though its parameters are optimized solely based on the task objective. We found that just averaging the predictions of different states (i.e., majority voting) does not give better answers.

## 6.3 ROBUSTNESS AND CALIBRATION

We test AKOrN's robustness to adversarial attacks and its uncertainty quantification performance on CIFAR10 and CIFAR10 with common corruptions (CC, Hendrycks & Dietterich (2019)). We train two types of networks: a convolutional AKOrN (*AKOrN*$^{\text{conv}}$) and AKOrN with both convolution and self-attention (*AKOrN*$^{\text{mix}}$). The former has three convolutional Kuramoto layers. The latter replaces the last block with an attentive Kuramoto block. We use AutoAttack (Caron et al., 2021) to evaluate the model's adversarial robustness.

***AKOrN*s are resilient against gradient-based attacks** The model is heavily regularized and achieves both good adversarial robustness and robustness to natural corruptions (Tab. 4). This is re-

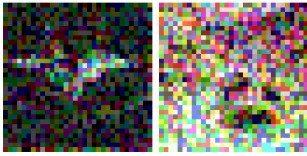 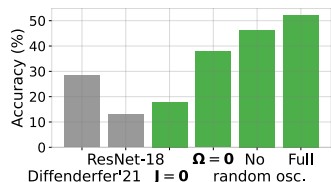

Figure 8: Robustness performance on random noise examples. Each bar plot shows classification accuracy on CIFAR10 with strong random noise ($\|\epsilon\|_\infty = 64/255$). The left two pictures are examples of images with that $\epsilon$. Green bars show accuracy when we ablate each element of *AKOrN*.

| Model | ↑ Accuracy | | | ↓ ECE |
|---|---|---|---|---|
| | Clean | Adv | CC | CC |
| Bartoldson et al. (2024) | 93.68 | 73.71 | 75.9 | 20.5 |
| Diffenderfer et al. (2021) | 96.56 | 0.00 | 92.8 | 4.8 |
| ViT | 91.44 | 0.00 | 81.0 | 9.6 |
| ResNet-18 | 94.41 | 0.00 | 81.5 | 8.9 |
| *AKOrN*$^{\text{conv}}$ | 88.91 | *58.91 | 83.0 | 1.3 |
| *AKOrN*$^{\text{mix}}$ | 91.23 | *51.56 | 86.4 | 1.4 |

Table 4: Robustness to adversarial examples by AutoAttack (Adv) and common corruptions (CC) on CIFAR10. *The attack is done by AutoAttack with EoT (Athalye et al., 2018). $\|\epsilon\|_\infty$ is set to 8/255. Expected Calibration Error (ECE) measures the alignment between confidence of the prediction and accuracy. The top two methods are selected from the highest-ranked methods on https://robustbench.github.io/.

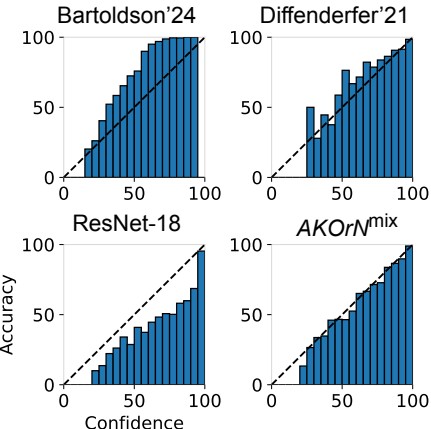

Figure 9: Confidence vs Accuracy plots on CIFAR10 with common corruptions.

markable, since conventional neural models need additional techniques such as adversarial training and/or adversarial purification to achieve good adversarial robustness. In contrast, *AKOrN* is robust by design, even when trained on only clean examples.

**K-Nets are well-calibrated and robust to strong random noise** We found that *AKOrN*s are robust to strong random noise (Fig. 8) and give good uncertainty estimation (on the bottom right in Fig. 9). Surprisingly, there is an almost perfect correlation between confidence and actual accuracy. This is similar to observations in generative models (Grathwohl et al., 2020; Jaini et al., 2024), where conditional generative models give well-calibrated outputs. Since *AKOrN*'s energy is not learned to model input distribution, we cannot tightly relate ours to such generative models. However, we speculate that *AKOrN*s' energy roughly approximates the likelihood of the input examples, and thus the oscillator state fluctuates according to the height of the energy, which would result in good calibration.

## 7 DISCUSSION & CONCLUSION

We propose *AKOrN*, which integrates the Kuramoto model into neural networks and scales to complex observations, such as natural images. *AKOrN*s learn object-binding features, can reason, and are robust to adversarial and natural perturbations with well-calibrated predictions. We believe our work provides a foundation for exploring a fundamental shift in the current neural network paradigm.

In the current formulation of *AKOrN*, each oscillator is constrained onto the sphere and each single oscillator cannot represent the 'presence' of the features like the rotating features in Löwe et al. (2023). Because of that, *AKOrN* would not perform well on memory tasks, where the model needs to remember the presence of events. This norm constraint also does not align with real biological neurons that have firing and non-firing states. Relaxing the hard norm constraint of the oscillator would be an interesting future direction in terms of both biological plausibility and applicability to a much wider range of tasks such as long-term temporal processing.

## 8 ACKNOWLEDGEMENT

Takeru Miyato and Andreas Geiger were supported by the ERC Starting Grant LEGO-3D (850533) and the DFG EXC number 2064/1 - project number 390727645. We thank Andy Keller, Lyle Muller, Terry Sejnowski, Bruno Olshausen, Christian Shewmake, Yue Song, Pietro Perona, Andreas Tolias, Masanori Koyama, Jun-nosuke Teramae, Bernhard Jaeger, Madhav Iyengar, and Daniel Dauner for their insightful feedback and comments. We thank Vladimir Fanaskov for providing more general proof of the Lyapunov property of our generalized Kuramoto model. Max Welling thanks the California Institute of Technology for hosting him in January and February 2024. Takeru Miyato acknowledges his affiliation with the ELLIS (European Laboratory for Learning and Intelligent Systems) PhD program.

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

# Appendix

## Table of Contents

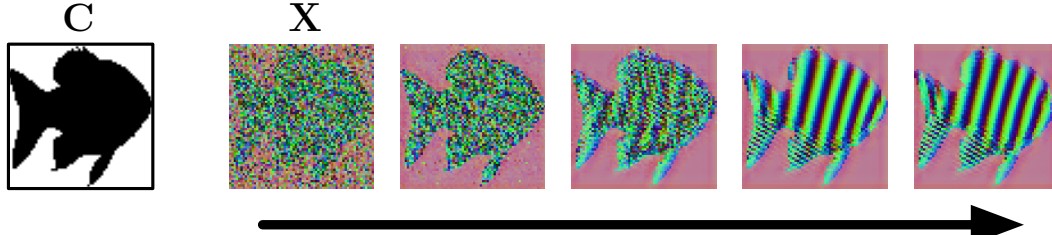

Figure 10: The transition of the 64×64 oscillator neurons ($N = 4$). (Left) Visualiaztion of $\mathbf{C}$. $\mathbf{c}_i$ on the white region is set to $\mathbf{1}$ and the black region is set to $\mathbf{0}$. (Right) Oscillators' time evolution. Similar colors indicate oscillators directing similar directions. The connectivity $\mathbf{J}$ is a $9 \times 9$ convolution kernel with random filters. The oscillators on the white region of $\mathbf{C}$ are aligned with the conditional stimuli and almost stay constant across time. The oscillators on the black region are largely influenced by the neighboring oscillators and exhibit wavy patterns.

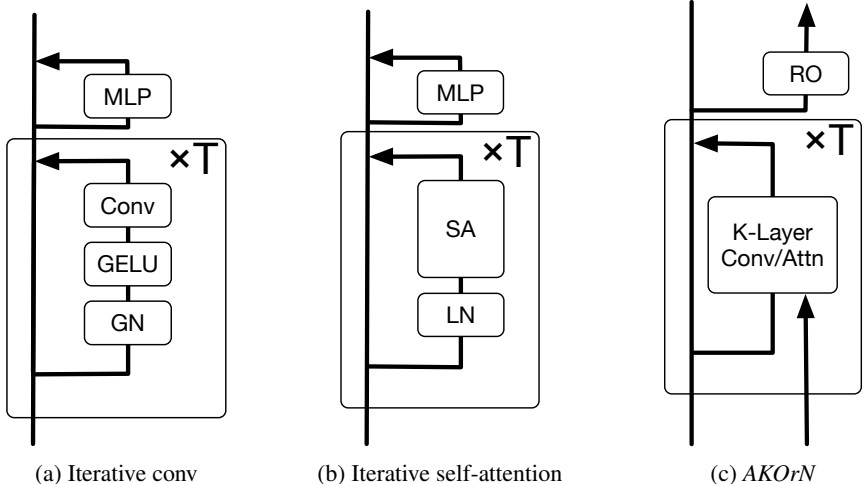

Figure 11: Block diagrams of (a) ItrConv (b) ItrSA, and (c) *AKOrN*. GN and LN stand for Group Normalization (Wu & He, 2018) and Layer normalization (Ba et al., 2016), respectively. The MLP in (a) or (b) is composed of a stack of GN or LN followed by Linear, GELU, and Linear layers. The hidden dim of MLP is set to $2\times$ (channel size). The number of heads in SA and the K-Layer with attentive connectivity is set to 8 throughout our experiments.

# A  MODEL ANALYSIS

In this section, we provide an extensive comparison of the architectural designs. Specifically, we show:

- A visual description of the projection operator and its effect on the performance (Sec. A.1)
- The Kuramoto model vs conventional residual update (Sec. A.2)
- The effect of the number of rotating dimensions $N$ (Sec. A.3)
- The efficacy of $\mathbf{C}$ and $\mathbf{m}$ in *AKOrN* (Sec. A.4).

Additionally, we show run-time comparisons between *AKOrN*s and their non-Kuramoto counterparts on different datasets in Sec. A.5.

## A.1  PROJECTION OPERATOR

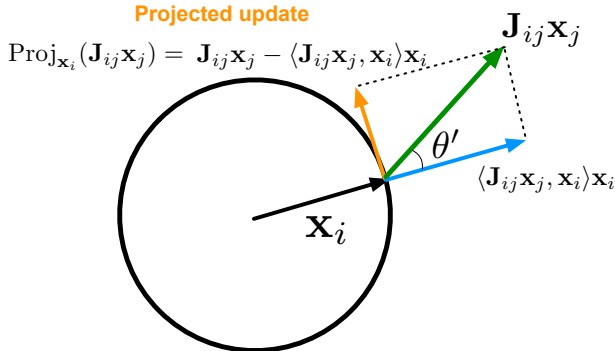

Figure 12: Visual description of $\mathrm{Proj}_{\mathbf{x}_i}(\mathbf{J}_{ij}\mathbf{x}_j)$. $\theta$' is the angle difference between $\mathbf{x}_i$ and $\mathbf{J}_{ij}\mathbf{x}_j$. Given the length of $\mathbf{J}_{ij}\mathbf{x}_j$, the length of the projected update in Eq. (2) and the negative energy in Eq. (3) are inversely proportional.

Fig. 12 illustrates a visual representation of the projection operator $\mathrm{Proj}_{\mathbf{x}_i}$ defined in Eq. (2). Note that this operator plays a key role in Riemannian optimization on the sphere, ensuring that the updated direction lies within the tangent space at the point $\mathbf{x}_i$ on the sphere.

**Relation between the vectorized Kuramoto model and the original one**  The vectorized Kuramoto model includes the original one in a special case. Suppose the case of $N = 2$, $\mathbf{c}_i = \mathbf{0}$, and having a scalar connection for $\mathbf{J}_{ij}$: $\mathbf{J}_{ij} = J_{ij}\mathbf{I}$ where $J_{ij} \in \mathbb{R}$ and $\mathbf{I}$ is the $2 \times 2$ identity matrix. Then we have $\theta' = \theta_j - \theta_i$ where $\theta_i, \theta_j = \arg(\mathbf{x}_i), \arg(\mathbf{x}_j)$. From the definition of trigonometric functions, we get

$$\langle \mathbf{J}_{ij}\mathbf{x}_j, \mathbf{x}_i \rangle \mathbf{x}_i = J_{ij}\cos(\theta_j - \theta_i)\mathbf{x}_i \tag{10}$$

$$\mathrm{Proj}_{\mathbf{x}_i}(\mathbf{x}_j) = \mathbf{J}_{ij}\mathbf{x}_j - \langle \mathbf{J}_{ij}\mathbf{x}_j, \mathbf{x}_i \rangle \mathbf{x}_i = J_{ij}\sin(\theta_j - \theta_i)\mathbf{x}_i^\perp, \tag{11}$$

where $\mathbf{x}_i^\perp$ is the unit vector perpendicular to $\mathbf{x}_i$ and its direction is increasing $\theta_i$. Thus the Eq. (2) is an extension of Eq. (1). This proof is just a rephrased version of Chandra et al. (2019) and Proposition 1 in Olfati-Saber (2006). Please refer to them for details.

Note that with or without Proj only changes the length of the update direction of each neuron. The updated $\mathbf{x}_i$ stays on the sphere since we normalize each updated neuron to be the unit vector in Eq. (5). We test *AKOrN* without Proj operators and summarize the results in Tab. 5. We see almost identical and a bit degraded performance on the CLEVR-TEx object discovery and Sudoku solving, respectively. Interestingly, without projection, the adversarial robustness and uncertainty quantification get worse than the original *AKOrN*.

| Proj$_\mathbf{x}$ | FG-ARI | MBO |
|---|---|---|
| ✗ | 79.0$\pm$2.5 | 56.2$\pm$1.0 |
| ✓ | 80.5$\pm$1.5 | 54.9$\pm$0.6 |

(a) CLEVR-Tex

| Proj$_\mathbf{x}$ | ID | OOD |
|---|---|---|
| ✗ | 99.9$\pm$0.0 | 45.0$\pm$1.9 |
| ✓ | **100.0**$\pm$0.0 | **51.7**$\pm$3.3 |

(b) Sudoku

| Proj$_\mathbf{x}$ | ↑ Accuracy Clean | Adv | CC | ↓ ECE CC |
|---|---|---|---|---|
| ✗ | **89.9** | 0.1 | **82.4** | 4.5 |
| ✓ | 84.6 | **64.9** | 78.3 | **1.8** |

(c) CIFAR10

Table 5: Ablation of Proj$_\mathbf{x}$.

## A.2 Replacing the Kuramoto model with the conventional residual update

Here, we conduct an ablation study of the Kuramoto updates. Specifically, we train a proposed *AKOrN* architecture on CLEVRTex and Sudoku, but without projection and normalization (Proj and Π in Eqs (4) and (5)). The update results in the conventional residual update. Tab. 6 shows that the ablated model degrades both the object discovery performance and Sudoku solving significantly, which clearly shows the large contribution of the Kuramoto update to the performances.

| Kuramoto | FG-ARI | MBO |
|---|---|---|
| ✗ | 66.2$\pm$1.6 | 51.4$\pm$0.1 |
| ✓ | **80.5**$\pm$1.5 | **54.9**$\pm$0.6 |

(a) CLEVR-Tex

| Kuramoto | ID | OOD |
|---|---|---|
| ✗ | 59.8$\pm$54.6 | 17.1$\pm$16.6 |
| ✓ | **100.0**$\pm$0.0 | **51.7**$\pm$3.3 |

(b) Sudoku

Table 6: Performance with or without the Kuramoto update rule.

## A.3 Number of rotating dimensions

Tab. 7 and Fig. 14 show *AKOrN* with $N = 2$, which is close to the original Kuramoto model as shown in A.1, significantly underfit in the object discovery experiments and the Sudoku solving. We do not observe significant improvement by increasing $N$ above 4 and a sudden drop when $N$ exceeds a certain number depending on datasets (See Fig. 16).

| $N$ | FG-ARI | MBO |
|---|---|---|
| 2 | 44.6$\pm$2.5 | 28.0$\pm$1.0 |
| 4 | **80.5**$\pm$1.5 | **54.9**$\pm$0.6 |

(a) CLEVR-Tex

| $N$ | ID | OOD |
|---|---|---|
| 2 | 0.3$\pm$0.4 | 0.0$\pm$0.0 |
| 4 | **100.0**$\pm$0.0 | **51.7**$\pm$3.3 |

(b) Sudoku

Table 7: The effect of the number of rotating dimensions. The inferior performance with $N = 2$ comes from the model's underfitting (See the next figures)

.

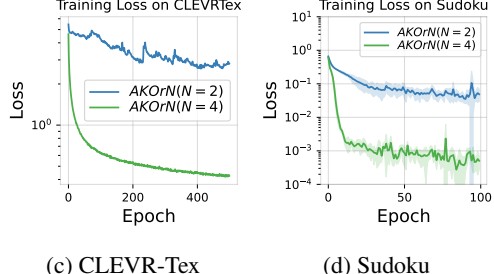

(c) CLEVR-Tex        (d) Sudoku

Figure 14: Loss curve comparison between models with $N = 2$ and $N = 4$.

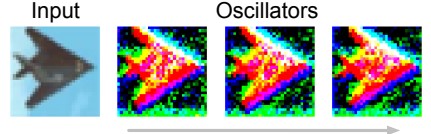

Figure 15: Noisy oscillators when $N = 2$. The three panels show the states of the oscillators at consecutive time steps. The animation file cifar10_block1.gif in the Supplementary Material provides more clear visualization of the fluctuations.

| Model | ↑ Accuracy Clean | Adv | CC | ↓ ECE CC |
|---|---|---|---|---|
| ResNet ($\sigma = 0.2$) | 85.2 | 22.3 | 75.1 | 2.3 |
| ResNet ($\sigma = 0.225$) | 83.9 | 25.5 | 73.6 | 2.6 |
| *AKOrN* ($N = 2$) | 84.6 | **64.9** | **78.3** | **1.8** |

Table 8: Robustness comparison with ResNets trained to resist Gaussian noises. $\sigma$ indicates the standard deviation of the noise added during training.

Figs 16 and 17 show the performance dependence on the choice of $N$. We here test the object discovery task and the sudoku solving. We observe a slight improvement as $N$ increases, followed by a sudden drop in performance in both tasks (except on the Shapes dataset in the object discovery task).

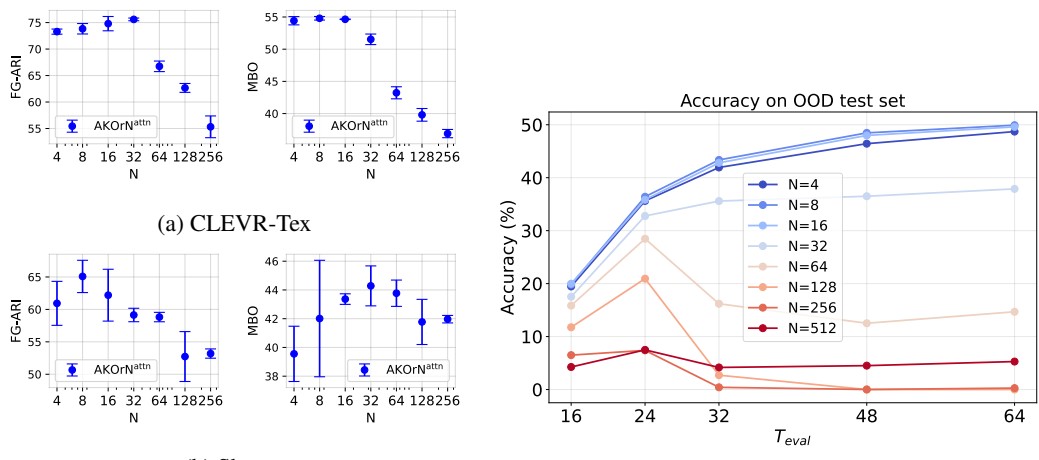

(a) CLEVR-Tex

(b) Shapes

Figure 16: FG-ARI and MBO vs oscillator dimensions $N$. Here, the models' channel $C$ is set to $256/N$. #layers and #(Kuramoto updates) are set to 1 and 8, respectively.

Figure 17: Sudoku board accuracy on the OOD test set. The models' channel $C$ is set to $512/N$.

### A.4 THE BIAS TERM $\mathbf{C}$ AND NORM-TAKING TERM $\mathbf{m}$

*AKOrN* employs a two-stream architecture to process observations, which helps stabilize training. Additionally, the norm-taking part $\mathbf{m}$ in Eq. (6) plays a key role in improving the model's fitness to the data. Fig. 18 presents a performance comparison with a model stacking only K-layers (Staking K-Layers) and *AKOrN* without $\mathbf{m}$. Here, 'Stacking K-Layers' removes the term $\mathbf{C}$ in Eq. (4) and instead processes an observation into $\mathbf{X}^{(0)}$ by using a single 3×3 convolution applied to the RGB input. We see the use of $\mathbf{C}$ and $\mathbf{m}$ significantly contributes to the loss minimization. The final test accuracies of Stacking K-Layers, *AKOrN* wo/ $\mathbf{m}$, and the original *AKOrN* are 62.9, 77.8, and 84.6, respectively.

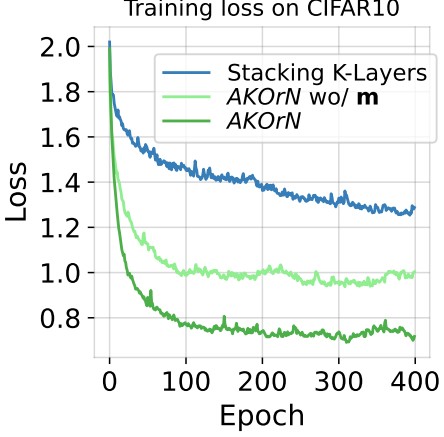

Figure 18: Effect of the use of $\mathbf{C}$ and $\mathbf{m}$ in Eq. (6)

## A.5 TRAINING & INFERENCE TIME

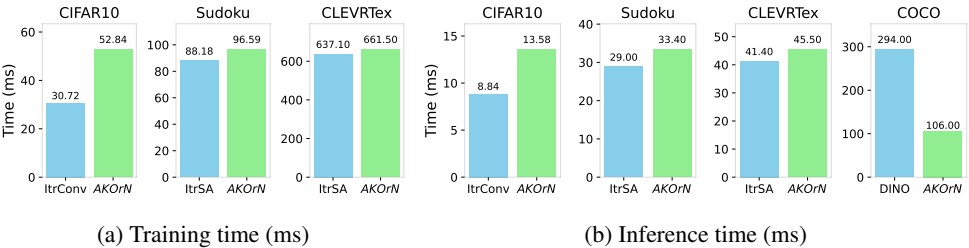

(a) Training time (ms)  (b) Inference time (ms)

Figure 19: Training and inference time in different tasks. Training time is the time taken to complete a single gradient step (excluding data loading). Inference time is the time taken for a single forward pass with a mini-batch size of 100.

# B ADDITIONAL DISCUSSION ON MOTIVATION AND RELATED WORK

## B.1 RELATION TO PHYSICS MODELS

Similar to how the Ising model is the basis for recurrent neural models, such as the Hopfield model (Hopfield, 1982), the Kuramoto model with symmetric lateral interactions can also be studied by viewing it as a model from statistical physics called the Heisenberg model (Mattis, 2012). In fact, we use more general version of the Kuramoto model which involves a symmetry-breaking term (akin to a magnetic field interaction) and asymmetric connections between the neurons. This not only is biologically plausible (synapses are not symmetric), it also leads to much better results in our experiments.

Non-equilibrium soft matter physics has studied models with nonreciprocal interactions, for instance in the field of "active matter". They have developed accurate coarse-grained hydrodynamics models to approximate the microscopic dynamics and observed very interesting behavior, such as symmetry-breaking phase transitions and resultant traveling waves representing so-called Goldstone modes (Fruchart et al., 2021). We hope that this opens the door to a deeper understanding of these models when employed as neural networks.

## B.2 RELATED WORKS ON THE NN ROBUSTNESS

Experimental proof of the conventional NNs' limited OOD generalization is represented by the vulnerability to adversarial examples (Szegedy et al., 2014; Goodfellow et al., 2014). The most effective way to resist such examples is training the model on adversarial examples generated by the model itself, which is called adversarial training (Goodfellow et al., 2014; Madry et al., 2017; Miyato et al., 2018; Zhang et al., 2019). Many other defenses have been proposed, but most of them were found to be not a fundamental solution (Tramer et al., 2020).

One framework that can produce more human-algined predictions is a generative classifier (Ng & Jordan, 2001; Bishop & Nasrabadi, 2006), where we train a model with both generative and discriminative objectives or turn a label conditional generative model into a discriminative model based on Bayes theorem. Interestingly, different generative classifiers trained with different methods share similar robust and calibration properties (Lee et al., 2017; Grathwohl et al., 2020; Li et al., 2023; Jaini et al., 2024). Generative classifiers are robust but involve costly generative training such as denoising diffusion (Li et al., 2023; Jaini et al., 2024), MCMC (Grathwohl et al., 2020) to generate negative samples, or unstable min-max optimization as GANs training (Lee et al., 2017). *AKOrN* shares similar robustness properties but without any generative objectives.

# C EXPERIMENTAL SETTINGS

We observe that both the readout module and conditional stimuli **C** are essential for stable training, especially when $N = 2$. We also see that *AKOrN* with $N = 2$ exhibits a strong regularity, which

| | Tetrominoes | dSprites | CLEVR | Shapes |
|---|---|---|---|---|
| |  |  |  |  |
| Training examples | 60,000 | 60,000 | 50,000 | 40,000 |
| Test examples | 320 | 320 | 320 | 1,000 |
| Image size | 32 | 64 | 128 | 40 |
| Max. #objects | 3 | 6 | 6 | 4 |
| Patch size | 4 | 4 | 8 | 2 |
| Patch resolution | 8 | 16 | 16 | 20 |
| Channel size | 128 | 128 | 256 | 256 |
| #internal steps ($T$) | 8 | 8 | 8 | 4 |
| #Epochs | 50 | 50 | 300 | 100 |
| Batchsize | | 256 | | |
| Learning rate | | 0.001 | | |
| Augmentations | | Random resize and crop + color jittering | | |
| #clusters set for eval | 4 | 7 | 11 | 5 |

Table 9: Experimental settings on Tetrominoes, dSprites, CLEVR, and Shapes.

acts positively on robustness performance while having negative effects on unsupervised object discovery and the Sudoku-solving experiments. We show results of *AKOrN* with $N = 4$ in those experiments. We do not observe significant improvement by increasing $N$ above 4 (See Fig. 16). Further experimental and mathematical analysis is needed to understand why this occurs, which could provide insights into how we can leverage both advantages.

Tabs 9-12 show experimental settings on each dataset (e.g. hyperparameters on models and optimization, the number of training and test examples, dataset statistics, etc...). For *AKOrN*, the channel size is set to (the channel size shown in the table)$/N$, so that the memory consumption and FLOPs are effectively the same between *AKOrN*s and their non-Kuramoto counterpart baselines. All models are trained with Adam (Kingma & Ba, 2015) without weight decay.

## C.1 UNSUPERVISED OBJECT DISCOVERY

We test on 4 synthetic benchmark datasets (Tetrominoes, dSprites, CLEVR, CLEVRTex), one synthetic dataset created by us (Shapes), and 2 real image benchmark datasets (PascalVOC, COCO2017). The Shapes dataset consists of images with 2–4 objects that are randomly sampled from four basic shapes (triangle, square, circle, and diamond). Note that each image can have multiple objects of the same shape together.

The kernel size of convolution layers in *AKOrN*$^{\text{conv}}$ and ItrConv is set to 5, 7, and 9 on Tetrominoes, dSprites, and CLEVR, respectively. In addition to ItrConv and ItrSA, we also train a ViT model (Dosovitskiy et al., 2021) as another baseline.

All networks process images similarly to ViT (Dosovitskiy et al., 2021). First, we patch each image into $H/P \times W/P$ patches where $H, W$ are the height and width of the image and $P$ is the patch size. We then apply the stack of blocks. The output of the final layer is further processed by global max-pooling followed by a single hidden layer MLP, whose output is used to compute the SimCLR loss. We used a conventional set of augmentations for SSL training: random resizing, cropping, and color jittering. We also apply horizontal flipping for the ImageNet pretraining. All models including baseline models have roughly the same number of parameters and are trained with shared hyperparameters such as learning rates and training epochs. See Tabs 9-11 for those hyperparameter details.

| | CLEVRTex | OOD | CAMO |
|---|---|---|---|
| |  |  |  |
| Training examples | 40,000 | - | - |
| Test examples | 5,000 | 10,000 | 2,000 |
| Image size | | 128 | |
| Max. #objects | | 10 | |
| Patch size | | 8 | |
| Patch resolution | | 16 | |
| Channel size | | 256 | |
| # internal steps ($T$) | | 8 | |
| # epochs | 500 | - | - |
| Batchsize | 256 | - | - |
| Learning rate | 0.0005 | - | - |
| Augmentations | Random resize and crop + color jittering | | |
| #clusters set for eval | | 11 | |

Table 10: Experimental settings on CLEVRTex and its variants (OOD, CAMO). We also train a large *AKOrN* model that is trained with the doubled channel size, and epochs. We denote that model by Large *AKOrN*.

In *AKOrN*, $\mathbf{C}^{(0)}$ is computed by the patched features of images, while each $\mathbf{x}_i$ is initialized by random oscillators sampled from the uniform distribution on the sphere. We use the identity function for $g$ in each readout module. In multi-block models, we apply Group Normalization (Wu & He, 2018) to $\mathbf{C}$ except for the last block's output $\mathbf{C}^{(L)}$.

For the Tetrominoes, dSprites, and CLEVR datasets, we train single-block models with $T = 8$. We observe that stacking multiple blocks does not yield improvements on those three datasets. On CLEVRTex, we train single- and two-block models with attentive connectivity and $T = 8$, while on ImageNet, we train a three-block *AKOrN* model with attentive connectivity and $T = 4$.

### C.1.1 METRICS

We use FG-ARI and MBO to evaluate cluster assignments, both of which are well-used metrics in object discovery tasks. Below are the summaries of the FG-ARI and MBO metrics.

- FG-ARI: The Adjusted Rand Index (ARI) computes how well the clusters align with object masks compared to random cluster assignments. The foreground ARI (FG-ARI) only considers foreground objects and is a well-used metric in object discovery tasks. The maximum value of 100 indicates perfect alignment between the obtained clusters and the object masks. If the cluster assignment is completely random or all features are assigned to the same cluster, the value is 0.

- MBO: The Mean Best Overlap (MBO) first assigns each cluster to the highest overlapping ground truth mask and then computes the average intersection-over-union (IoU) of all pairs. The value takes 100 at maximum. Following the literature, we exclude the background mask from the MBO evaluation. Since MBO computes IoU, tightly aligned object masks give a higher value than FG-ARI (FG-ARI does not penalize the mask extending into the background region).

| | ImageNet | PascalVOC | COCO2017 |
|---|---|---|---|
| |  |  |  |
| Training examples | 1,281,167 | - | - |
| Test examples | - | 1,449 | 5,000 |
| Image size | 256 | 256 | 256 |
| Patch size | | 16 | |
| Patch resolution | | 16 | |
| Channel size | | 768 | |
| # Blocks | | 3 | |
| # internal steps ($T$) | | 4 | |
| # epochs | 400 | - | - |
| Batchsize | 512 | - | - |
| Learning rate | 0.0005 | - | - |
| #clusters set for eval | - | 4 | 7 |

Table 11: Experimental settings on ImageNet pratraining and on the PascalVOC and COCO2017 evaluation. For SimCLR training augmentations, we use random resize and crop, color jittering, and horizontal flipping.

| | Sudoku(ID) (Wang et al., 2019) | Sudoku(OOD) (Palm et al., 2018) |
|---|---|---|
| |  |  |
| Training examples | 9,000 | - |
| Test examples | 1,000 | 18,000 |
| Channel size | | 512 |
| # epochs | 100 | - |
| Batchsize | 100 | - |
| Learning rate | 0.0005 | - |

Table 12: Sudoku puzzle datasets.

Figure 20: $2\times$ *up-tiling*. First, we create horizontally or/and vertically shifted images with stride equal to (patchsize/2) and compute the model's output on each shifted image. We then interleave each token feature to make a $2\times$ upsampled feature map.

### C.1.2 UPSAMPLE FEATURES BY UP-TILING

When we compute the cluster assignment, we upsample the output features by *up-tiling*. In this approach, the model processes a set of slightly shifted versions of the input image along the horizontal and/or vertical axes. These shifted images allow the model to generate slightly different feature maps for each shifted position. The final higher-resolution feature map is then created by interleaving these feature maps, effectively combining the shifted perspectives into a single, more detailed representation. This up-tiling enables us to get finer cluster assignments and substantially improves the object discovery performance of our *AKOrN*. We show a pictorial explanation in Fig. 20 and PyTorch code in Code 1. In Fig. 21, we compare up-tiled features with the original features and features with bilinear upsampling. Fig. 22 shows some examples of up-tiled features. We apply up-tiling with the scale factor of 4 for producing numbers on Tabs 1 and 2 as well as for cluster visualization in Figs 4,5 and Figs 31-36. Unless otherwise stated, no upsampling is performed when computing the cluster assignment.

Code 1: PyTorch code for up-tiling

```python
def create_shifted_imgs(img, psize, stride):
    H, W = img.shape[-2:]
    img = F.interpolate(img,
                        (H+psize-stride, W+psize-stride),
                        mode='bilinear', align_corners=False)
    imgs = []
    for h in range(0, psize, stride):
        for w in range(0, psize, stride):
            imgs.append(img[:, :, h:h+H, w:w+W])
    return imgs

def uptiling(model, images, psize=16, s=2):
    """
    Args:
        model: a function that takes [B,C,H,W]-shaped tensor
               and outputs [B,C,H/psize,W/psize]-shaped tensor.
        images: a tensor of shape [B, C, H, W].
        psize: the patch size of the model.
        s: scale factor. The resulting features will
               be upscaled to [s*H/psize, s*W/psize]
               where (H, W) are the original image size.
               Must be equal to or less than the patch size.
    Returns:
        nimgs: a tensor of shape [B, C, s*H/psize, s*W/psize]
    """
    B = images.shape[0]
    stride = psize // s
    # Create shifted images.
    shifted_imgs = create_shifted_imgs(images, psize, stride)
    # Compute a feature map on each shifted image.
    outputs = []
    for i in range(len(shifted_imgs)):
        with torch.no_grad():
            output = model(shifted_imgs[i].cuda())
            outputs.append(output.detach().cpu())
    # Tile the output feature maps.
    oh, ow = outputs[0].shape[-2:]
    nimgs = torch.zeros(B, outputs[0].shape[1], oh, s, ow, s)
    for h in range(s):
        for w in range(s):
            nimgs[:, :, :, h, :, w] = outputs[h*s+w]
    # Reshape into [B, C, s*(H/psize), s*(W/psize)]
    nimgs = nimgs.view(, -1, oh*nh, ow*nw)
    return nimgs
```

## C.2 SUDOKU SOLVING

The task is to fill a 9×9 grid, given some initial digits from 1 to 9, so that each row, column, and 3×3 subgrid contains all digits from 1 to 9. While the task may be straightforward if the game's rules are known, the model must learn these rules solely from the training set. Example boards are shown in Tab. 12.

We train *AKOrN* with attentive connections, the ItrSA model, and a conventional transformer model. We denote them by *AKOrN*$^{\text{attn}}$, ItrSA, and Transformer, respectively. *AKOrN*$^{\text{attn}}$ has almost the same architecture used in the object discovery task except for $g$ in the readout module, which is composed of the norm computation layer followed by a stack of ReLU, and linear layer.

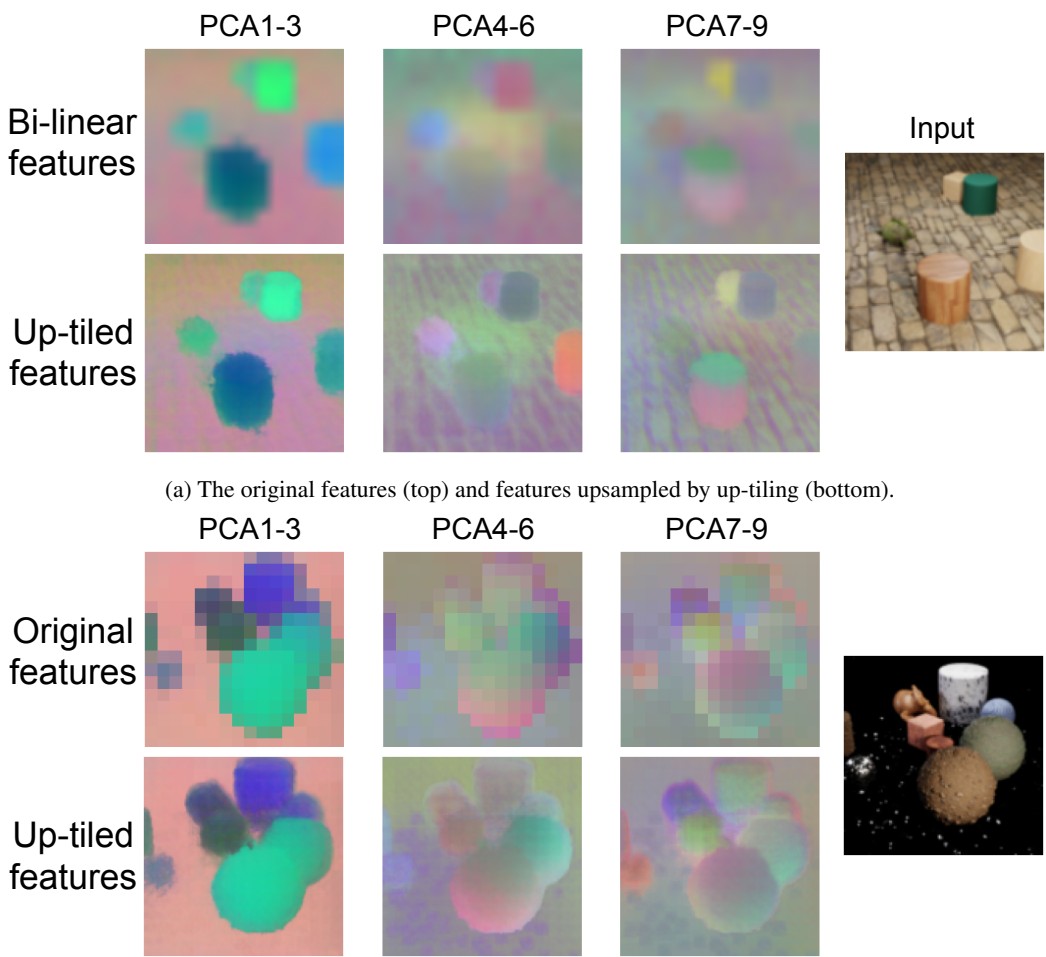

(a) The original features (top) and features upsampled by up-tiling (bottom).

(b) Features upsampled by bilinear upsampling (top) and by up-tiling (bottom).

Figure 21: Comparison of *AKOrN*'s output features upsampled by different methods. PCA$\{i - j\}$ indicates that the corresponding column's panels represent the features' $i$-th to $j$-th PCA components. The scaling factor of up-tiling is set to 8.

The input for each model is 9×9 digits from 0 to 9 (0 for blank, 1-9 for given digits). We first embed each digit into a 512-dimensional token vector. The 9×9 tokens are then flattened into 81 tokens. We apply each model to this token sequence and compute the prediction on each square by applying the softmax layer to each output token of the final block. All models are trained to minimize cross-entropy loss for 100 epochs.

The number of blocks of both ItrSA and *AKOrN* is set to one. We tested models with more than one block but found no improvement on the ID test set and a decline in OOD performance. Similar to the object discovery experiments, a transformer results in even worse performance than the ItrSA model (Tab. 19).

## C.3 ROBUSTNESS AND CALIBRATION ON CIFAR10

We train two types of networks for this task: a convolution-based *AKOrN* and *AKOrN* with a combination of convolution and attention. The former has three proposed blocks, and all of the Kuramoto layer's connectivities are convolutional connectivity. The kernel sizes are 9,7, and 5 from shallow to deep, and $T$ is set to 3 for all blocks. Between consecutive blocks, a single convolution with a stride being 2 is applied to each of $\mathbf{C}$ and $\mathbf{X}$. Thus, the feature resolution of the final block's output is $8 \times 8$. Each readout module's $g$ is Batch Normalization (Ioffe & Szegedy, 2015) followed by ReLU,

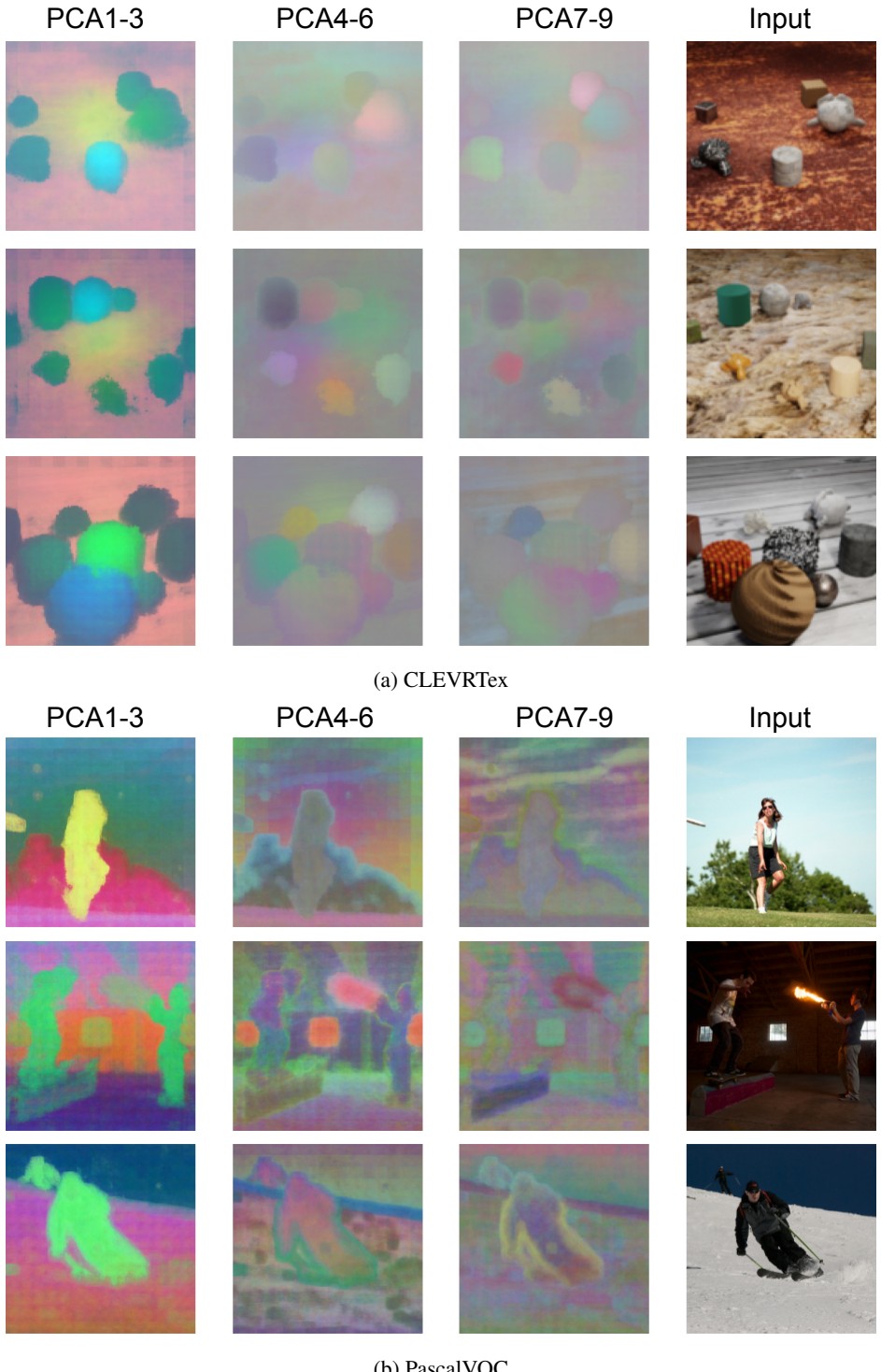

Figure 22: Up-tilied feature maps on CLEVRTex and PascalVOC. The scale factors are set to 8 and 16 for CLEVRTex and PascalVOC, respectively.

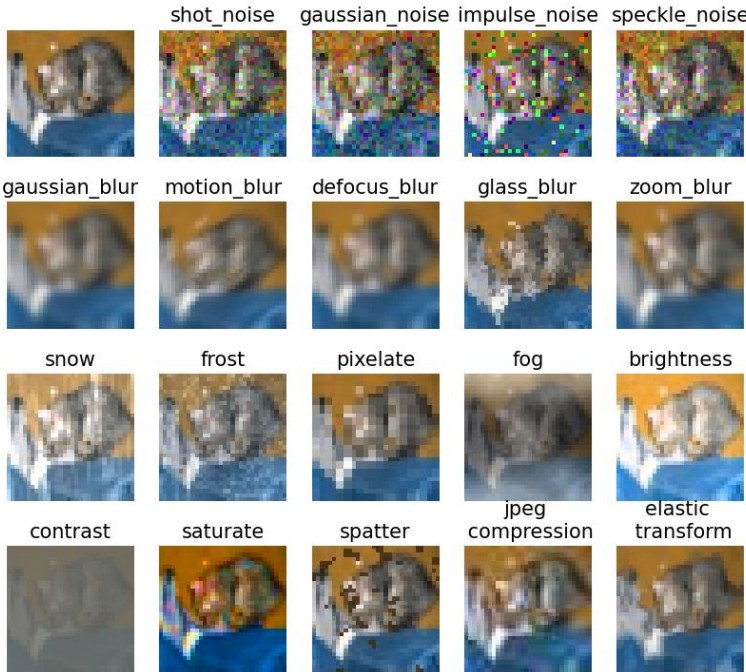

Figure 23: Example images in the Common Corruption dataset (CIFAR10-C). The top right image is the original clean image.

and a 3×3 convolution. $\mathbf{C}^{(3)}$ is average-pooled followed by the softmax layer that makes category predictions. The latter network is identical to the former one except for the third block, which we replace with the block with attentive connectivity. For this attentive model, different timesteps $T$ are set across different blocks, which are $[6, 4, 2]$ from shallow to deep.

For ResNet-18 and *AKOrN*, we first conduct pre-training on the Tiny-imagenet (Le & Yang, 2015) dataset with the SimCLR loss for 50 epochs with batchsize of 512. We observe that this pre-training is effective for *AKOrN* and improves the CIFAR10 clean accuracy compared to training from scratch (from 87% to 91%). The ImageNet pretraining slightly improves ResNet's clean accuracy (from 94.1% to 94.4%). Each model is then trained on CIFAR10 for 400 epochs. We apply augmentations, including random scaling and cropping, color jittering, and horizontal flipping, along with AugMix (Hendrycks et al., 2020), as commonly used in robustness benchmarks. Both models are trained to minimize the cross-entropy loss.

We also train an ItrConv model as a non-Kuramoto counterpart for this robustness experiment. To construct the ItrConv model, We replace each block of $AKOrN^{\mathrm{conv}}$ with the ItrConv block shown in Fig. 11 and set the same kernel size to each layer as $AKOrN^{\mathrm{conv}}$ (i.e. 9, 7, and 5 from shallow to deep layers). Hyperparameters such as the number of channels, learning rate, and others are shared with $AKOrN^{\mathrm{conv}}$.

# D    ADDITIONAL EXPERIMENTAL RESULTS

## D.1    POSITIONAL ENCODING FOR THE ATTENTIVE CONNECTIVITY

We need a positional encoding (PE) for *AKOrN* with attentive connectivity. We found GTA-type PE (Miyato et al., 2024) is effective and used for *AKOrN* throughout our experiments. Comparison to absolute positional encoding (APE) (Vaswani et al., 2017) and RoPE (Su et al., 2021) is shown in Tab. 13. GTA does not improve the baseline ItrSA models.

|  |  | CLEVRTex | |
|--|--|--|--|
|  | PE | FG-ARI | MBO |
| ItrSA | APE | 66.9 | 42.2 |
|  | GTA | 66.1 | 43.4 |
| AKOrN | APE | 72.0 | 51.4 |
|  | RoPE | 65.7 | 50.2 |
|  | GTA | 75.6 | 57.7 |

(a) CLEVRTex

|  | PE | Sudoku(OOD) |
|--|--|--|
| ItrSA | APE | $34.4_{\pm5.4}$ |
|  | GTA | $24.3_{\pm7.8}$ |
| AKOrN | APE | $48.1_{\pm9.1}$ |
|  | RoPE | $48.4_{\pm5.6}$ |
|  | GTA | $51.7_{\pm3.3}$ |

(b) Sudoku (OOD Test)

Table 13: Comparison of positional encoding schemes. The number of blocks is one for all models. The Sudoku results of AKOrNs are obtained with test-time extensions of the Kuramoto steps ($T_{\text{eval}} = 128$) but without the energy-based voting.

## D.2 UNSUPERVISED OBJECT DISCOVERY

### D.2.1 MBO ON THE SYNTHETIC DATASETS

Fig 24 shows *AKOrN*$^{\text{conv}}$ and *AKOrN*$^{\text{attn}}$ outperform their counterparts on almost every dataset in terms of MBO, as well as FG-ARI shown in Fig 3.

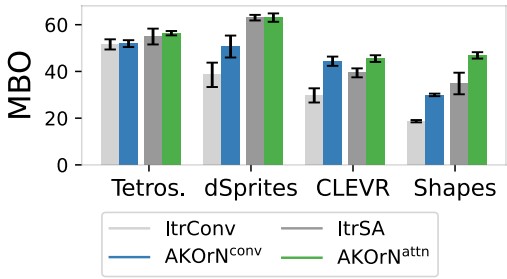

Figure 24: MBO on Tetrominoes, dSprites, CLEVR, and Shapes.

### D.2.2 MBO$_i$ VS # CLUSTERS

Fig. 25 shows AKOrN outperforms the other SSL methods across a wide range of the numbers of clusters, demonstrating AKOrN's robustness to variations in the number of clusters on object discovery performance.

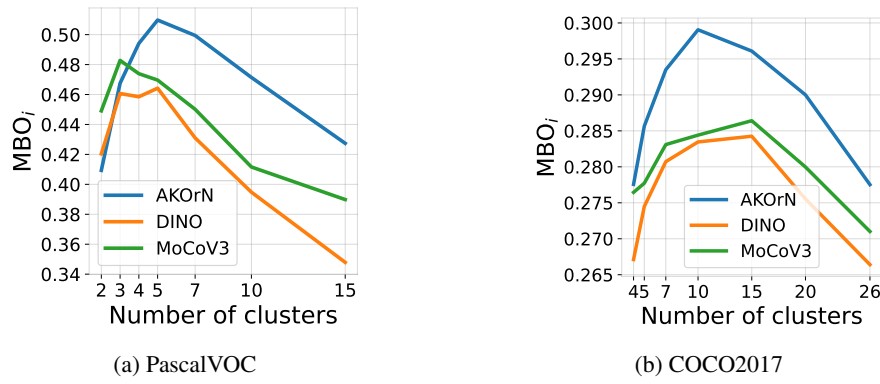

(a) PascalVOC

(b) COCO2017

Figure 25: MBO$_i$ vs the number of clusters used for evaluation.

### D.2.3 FULL TABLES OF OBJECT DISCOVERY PERFORMANCE

Tabs 14-18 show extended comparisons between *AKOrN* and existing models.

| Model | Tetrominoes | | dSprites | | CLEVR | |
|---|---|---|---|---|---|---|
| | FG-ARI | MBO | FG-ARI | MBO | FG-ARI | MBO |
| ItrConv | $59.0_{\pm2.9}$ | $51.6_{\pm2.2}$ | $29.1_{\pm6.2}$ | $38.5_{\pm5.2}$ | $49.3_{\pm5.1}$ | $29.7_{\pm3.0}$ |
| *AKOrN*$^{\mathrm{conv}}$ | $76.4_{\pm0.8}$ | $51.9_{\pm1.5}$ | $63.8_{\pm7.7}$ | $50.7_{\pm4.7}$ | $59.0_{\pm4.3}$ | $44.4_{\pm2.0}$ |
| ItrSA | $85.8_{\pm0.8}$ | $54.9_{\pm3.4}$ | $68.1_{\pm1.4}$ | $63.0_{\pm1.2}$ | $82.5_{\pm1.7}$ | $39.4_{\pm1.9}$ |
| *AKOrN*$^{\mathrm{attn}}$ | $88.6_{\pm1.7}$ | $56.4_{\pm0.9}$ | $78.3_{\pm1.3}$ | $63.0_{\pm1.8}$ | $91.0_{\pm0.5}$ | $45.5_{\pm1.4}$ |
| (+up-tiling ($\times4$)) | | | | | | |
| *AKOrN*$^{\mathrm{attn}}$ | $93.1_{\pm0.3}$ | $56.3_{\pm0.0}$ | $87.1_{\pm1.0}$ | $60.2_{\pm1.9}$ | $94.6_{\pm0.7}$ | $44.7_{\pm0.7}$ |
| (Synchrony-based models) | | | | | | |
| CAE (Löwe et al., 2022) | $78_{\pm7}$ | - | $51_{\pm8}$ | - | $27_{\pm13}$ | - |
| CtCAE (Stanić et al., 2023) | $84_{\pm9}$ | - | $56_{\pm11}$ | - | $54_{\pm2}$ | - |
| SynCx (Gopalakrishnan et al., 2024) | $89_{\pm1}$ | - | $82_{\pm1}$ | - | $59_{\pm3}$ | - |
| Rotating Features (Löwe et al., 2023) | $42_{\pm9}$ | - | $88.8_{\pm1.5}$ | $86.3_{\pm1.1}$ | $66.4_{\pm1.3}$ | $60.8_{\pm1.7}$ |
| (Slot-based model) | | | | | | |
| Slot-Attnetion (Locatello et al., 2020) | $99.5_{\pm0.2}$ | - | $91.3_{\pm0.3}$ | - | $98.8_{\pm0.3}$ | - |

Table 14: Object discovery results on synthetic datasets. We show the mean and std of the metrics of models with 3 different random seeds for the weight initialization.

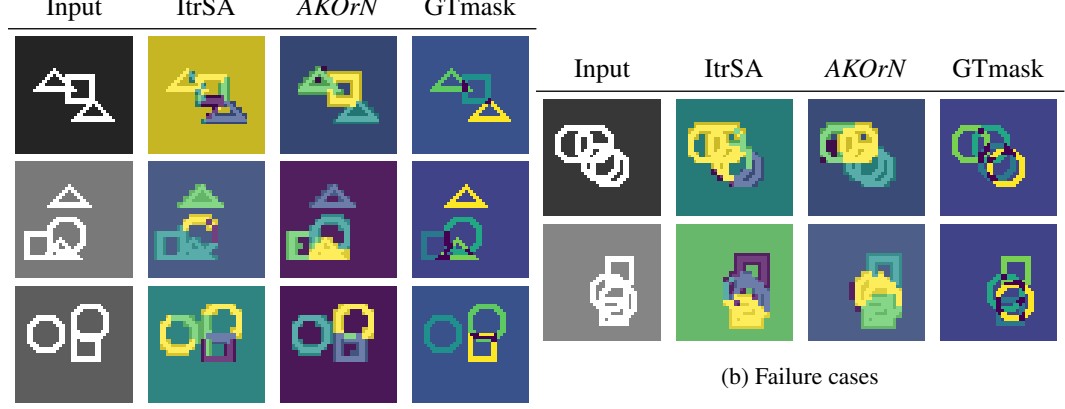

(a) Comparision between ItrSA and *AKOrN*

(b) Failure cases

Figure 26: Cluster visualization on Shapes. (b) Both ItrSA and *AKOrN* sometimes fail at separating overlapping objects with complex configurations.

| Model | FG-ARI | MBO |
|---|---|---|
| ItrConv | $21.0_{\pm0.8}$ | $18.7_{\pm0.5}$ |
| *AKOrN*$^{\mathrm{conv}}$ | $\mathbf{46.6}_{\pm2.1}$ | $\mathbf{30.0}_{\pm0.6}$ |
| ItrSA | $57.0_{\pm5.5}$ | $34.8_{\pm4.6}$ |
| *AKOrN*$^{\mathrm{attn}}$ | $\mathbf{71.3}_{\pm4.2}$ | $\mathbf{46.9}_{\pm1.4}$ |

Table 15: Object discovery performance on Shapes. We show the mean and std of the metrics of models with 3 different random seeds for the weight initialization.

| | $L$ | | |
|---|---|---|---|
| Model | 1 | 2 | 3 |
| ItrSA | $48.9_{\pm1.0}$ | $49.3_{\pm2.5}$ | $57.0_{\pm5.5}$ |
| *AKOrN*$^{\mathrm{attn}}$ | $\mathbf{52.5}_{\pm4.8}$ | $\mathbf{65.5}_{\pm2.0}$ | $\mathbf{71.3}_{\pm4.2}$ |

(c) FG-ARI

| | $L$ | | |
|---|---|---|---|
| Model | 1 | 2 | 3 |
| ItrSA | $38.8_{\pm1.3}$ | $37.2_{\pm2.1}$ | $34.8_{\pm4.6}$ |
| *AKOrN*$^{\mathrm{attn}}$ | $\mathbf{40.2}_{\pm2.1}$ | $\mathbf{44.6}_{\pm1.2}$ | $\mathbf{46.9}_{\pm1.4}$ |

(d) MBO

Table 16: Object discovery performance on Shapes varying the number of layers $L$.

| Model | CLEVRTex | | OOD | | CAMO | |
|---|---|---|---|---|---|---|
| | FG-ARI | MBO | FG-ARI | MBO | FG-ARI | MBO |
| ViT ($L=8, T=1$) | $46.4_{\pm 0.6}$ | $25.1_{\pm 0.7}$ | $44.1_{\pm 0.5}$ | $27.2_{\pm 0.5}$ | $32.5_{\pm 0.6}$ | $16.1_{\pm 1.1}$ |
| ItrSA ($L=1$) | $65.7_{\pm 0.3}$ | $44.6_{\pm 0.9}$ | $64.6_{\pm 0.8}$ | $45.1_{\pm 0.4}$ | $49.0_{\pm 0.7}$ | $30.2_{\pm 0.8}$ |
| ItrSA ($L=2$) | $76.3_{\pm 0.4}$ | $48.5_{\pm 0.1}$ | $74.9_{\pm 0.8}$ | $46.4_{\pm 0.5}$ | $61.9_{\pm 1.3}$ | $37.1_{\pm 0.5}$ |
| *AKOrN*$^{\mathrm{attn}}$ ($L=1$) | $75.6_{\pm 0.2}$ | $55.0_{\pm 0.0}$ | $73.4_{\pm 0.4}$ | $56.1_{\pm 1.1}$ | $59.9_{\pm 0.1}$ | $44.3_{\pm 0.9}$ |
| *AKOrN*$^{\mathrm{attn}}$ ($L=2$) | $80.5_{\pm 1.5}$ | $54.9_{\pm 0.6}$ | $79.2_{\pm 1.2}$ | $55.7_{\pm 0.5}$ | $67.7_{\pm 1.5}$ | $46.2_{\pm 0.9}$ |
| (+up-tiling ($\times 4$)) | | | | | | |
| *AKOrN*$^{\mathrm{attn}}$ ($L=2$) | $87.7_{\pm 1.0}$ | $55.3_{\pm 2.1}$ | $85.2_{\pm 0.9}$ | $55.6_{\pm 1.5}$ | $74.5_{\pm 1.2}$ | $45.6_{\pm 3.4}$ |
| Large *AKOrN*$^{\mathrm{attn}}$ ($L=2$) | $88.5_{\pm 0.9}$ | $59.7_{\pm 0.9}$ | $87.7_{\pm 0.3}$ | $60.8_{\pm 0.6}$ | $77.0_{\pm 0.5}$ | $53.4_{\pm 0.7}$ |
| [*]MONet (Burgess et al., 2019) | $19.8_{\pm 1.0}$ | - | $37.3_{\pm 1.0}$ | - | $31.5_{\pm 0.9}$ | - |
| [†]SLATE (Singh et al., 2022) | $44.2_{\pm NA}$ | $50.9_{\pm NA}$ | - | - | - | - |
| [*]Slot-Attetion (Locatello et al., 2020) | $62.4_{\pm 2.3}$ | - | $58.5_{\pm 1.9}$ | - | $57.5_{\pm 1.0}$ | - |
| Slot-diffusion (Wu et al., 2023) | $69.7_{\pm NA}$ | $61.9_{\pm NA}$ | - | - | - | - |
| [†]SLATE$^+$ (Singh et al., 2022) | $70.7_{\pm NA}$ | $54.9_{\pm NA}$ | - | - | - | - |
| [†]LSD (Jiang et al., 2023) | $76.4_{\pm NA}$ | $72.4_{\pm NA}$ | - | - | - | - |
| Slot-diffusion+BO (Wu et al., 2023) | $78.5_{\pm NA}$ | $68.7_{\pm NA}$ | - | - | - | - |
| [*]DTI (Monnier et al., 2021) | $79.9_{\pm 1.37}$ | - | $73.7_{\pm 1.0}$ | - | $72.9_{\pm 1.9}$ | - |
| [*]I-SA (Chang et al., 2022) | $79.0_{\pm 3.9}$ | - | $83.7_{\pm 0.9}$ | - | $57.2_{\pm 13.3}$ | - |
| BO-SA (Jia et al., 2023) | $80.5_{\pm 2.5}$ | - | $86.5_{\pm 0.2}$ | - | $63.7_{\pm 6.1}$ | - |
| [‡]NSI (Dedhia & Jha, 2024) | $89.9_{\pm 0.0}$ | $46.6_{\pm 0.0}$ | - | - | - | - |
| ISA-TS (Biza et al., 2023) | $92.9_{\pm 0.4}$ | - | $84.4_{\pm 0.8}$ | - | $86.2_{\pm 0.8}$ | - |
| [†]Jung et al. (2024) | $93.1_{\pm NA}$ | $75.4_{\pm NA}$ | - | - | - | - |
| [p]Sauvalle & de La Fortelle (2023) | $94.8_{\pm 0.5}$ | - | $83.1_{\pm 0.8}$ | - | $87.3_{\pm 3.8}$ | - |

Table 17: Object discovery on CLEVRTex (Karazija et al., 2021). [†]Use Openimages (Kuznetsova et al., 2020)-pretrained encoder. Numbers are from Jung et al. (2024). [‡]Use ImageNet-pretrained DINO. [*]Numbers taken from Jia et al. (2023). [p]Use Imagenet-pretrained backbone models. We show the mean and std of the metrics of models with 3 different random seeds for the weight initialization.

| Model | PascalVOC MBO$_i$ | PascalVOC MBO$_c$ | COCO2017 MBO$_i$ | COCO2017 MBO$_c$ |
|---|---|---|---|---|
| (slot-based models) | | | | |
| Slot-attention (Locatello et al., 2020) | 22.2 | 23.7 | 24.6 | 24.9 |
| SLATE (Singh et al., 2021) | 35.9 | 41.5 | 29.1 | 33.6 |
| (DINO + synchrony-based models) | | | | |
| Rotating Features (Löwe et al., 2023) | 40.7 | 46.0 | - | - |
| (DINO + slot-based model) | | | | |
| NSI (Dedhia & Jha, 2024) | - | - | 28.1 | 32.1 |
| DINOSAUR (Seitzer et al., 2023) | 44.0 | 51.2 | 31.6 | 39.7 |
| Slot-diffusion (Wu et al., 2023) | 50.4 | 55.3 | 31.0 | 35.0 |
| SPOT (Kakogeorgiou et al., 2024) | 48.3 | 55.6 | **35.0** | **44.7** |
| (SSL models) | | | | |
| MAE (He et al., 2022) | 33.8 | 37.7 | 22.9 | 28.3 |
| DINO (Caron et al., 2021) | 44.3 | 50.0 | 28.8 | 35.8 |
| MoCoV3 (Chen et al., 2021a) | 47.3 | 53.0 | 28.7 | 36.0 |
| *AKOrN*$^{\text{attn}}$ | 50.3 | 58.2 | 30.2 | 38.2 |
| (SSL models + up-tiling ($\times 4$)) | | | | |
| MAE | 34.0 | 38.3 | 23.1 | 28.5 |
| DINO | 47.2 | 53.5 | 29.4 | 37.0 |
| MoCoV3 | 44.6 | 50.5 | 29.0 | 35.9 |
| *AKOrN*$^{\text{attn}}$ | **52.0** | **60.3** | 31.3 | 40.3 |

Table 18: Object discovery on PascalVOC and COCO2017.

### D.2.4 TRAINING EPOCHS VS MBO

Fig. 27 shows that MBO$_i$ and MBO$_c$ scores on Pascal and COCO improve as ImageNet pretraining progresses. Similar observations are made on CLEVRTex datasets, where larger AKOrNs give better object discovery performance (see Figs 32-34 and Tab. 17). These results indicate that there is an alignment between the SSL training with *AKOrN* and learning object-binding features and that increasing parameters and computational resources can further enhance the object discovery performance.

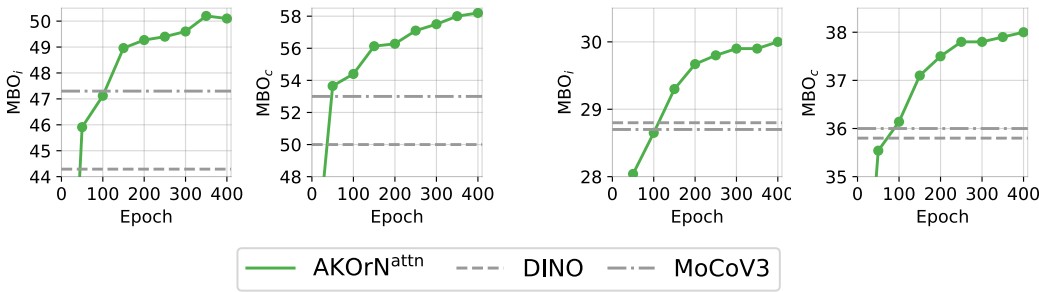

Figure 27: MBO$_i$ and MBO$_c$ vs. training epochs. (Left) PascalVOC (Right) COCO2017.

## D.3 SUDOKU SOLVING

### D.3.1 FULL TABLE OF BOARD ACCURACY

| Model | ID | OOD |
|---|---|---|
| Energy Transformer (Hoover et al., 2023) | $1.0_{\pm 1.0}$ | $0.0_{\pm 0.0}$ |
| *Symmetrized *AKOrN* ($L = 1, T = 16$) | $84.6_{\pm 14.2}$ | $1.4_{\pm 1.7}$ |
| Transformer | $98.6_{\pm 0.3}$ | $5.2_{\pm 0.2}$ |
| ItrSA ($L = 1, T = 16$) | $99.7_{\pm 0.3}$ | $14.1_{\pm 2.7}$ |
| *AKOrN*$^{\mathrm{attn}}$ wo $\Omega$ ($L = 1, T = 16$) | $99.8_{\pm 0.1}$ | $16.6_{\pm 2.2}$ |
| *AKOrN*$^{\mathrm{attn}}$ ($L = 1, T = 16$) | $99.8_{\pm 0.1}$ | $16.6_{\pm 2.1}$ |
| (+Test time extensions of internal steps) | | |
| ItrSA ($T_{\mathrm{eval}} = 32$) | $95.7_{\pm 8.5}$ | $34.4_{\pm 5.4}$ |
| *AKOrN*$^{\mathrm{attn}}$ wo $\Omega$ ($T_{\mathrm{eval}} = 128$) | $\mathbf{100.0}_{\pm 0.0}$ | $49.6_{\pm 3.3}$ |
| *AKOrN*$^{\mathrm{attn}}$ ($T_{\mathrm{eval}} = 128$) | $\mathbf{100.0}_{\pm 0.0}$ | $51.7_{\pm 3.3}$ |
| ($T_{\mathrm{eval}} = 128$, Energy-based voting ($K = 100$)) | | |
| *AKOrN*$^{\mathrm{attn}}$ wo $\Omega$, $E_{T_{\mathrm{eval}}}$ | $\mathbf{100.0}_{\pm 0.0}$ | $46.8_{\pm 9.0}$ |
| *AKOrN*$^{\mathrm{attn}}$, $E_{T_{\mathrm{eval}}}$ | $\mathbf{100.0}_{\pm 0.0}$ | $74.0_{\pm 5.6}$ |
| *AKOrN*$^{\mathrm{attn}}$, $\sum_t E_t$ | $\mathbf{100.0}_{\pm 0.0}$ | $81.6_{\pm 1.5}$ |
| ($T_{\mathrm{eval}} = 512$, Energy-based voting ($K = 1000$)) | | |
| *AKOrN*$^{\mathrm{attn}}$, $\sum_t E_t$ | $\mathbf{100.0}_{\pm 0.0}$ | $\mathbf{89.5}_{\pm 2.5}$ |
| SAT-Net (Wang et al., 2019) | 98.3 | 3.2 |
| Diffusion (Du et al., 2024) | 66.1 | 10.3 |
| IREM (Du et al., 2022) | 93.5 | 24.6 |
| RRN (Palm et al., 2018) | 99.8 | 28.6 |
| R-Transformer (Yang et al., 2023) | **100.0** | 30.3 |
| IRED (Du et al., 2024) | 99.4 | 62.1 |

Table 19: Board accuracy on Sudoku Puzzles. The harder dataset (OOD) has fewer conditional digits per example than the train set (17-34 in the harder dataset while 31-42 in the train set). We show the mean and std of the accuracy of models with different random seeds for the weight initialization. *Numbers are calculated with excluding one trained model that has stuck during training. For energy-based voting, we found the sum of energy values across timesteps ($\sum_t E_t$) indicate board correctness better than the energy at the last time step ($E_{T_{\mathrm{eval}}}$).

### D.3.2 EFFECT OF THE NATURAL FREQUENCY TERM IN ENERGY-BASED VOTING

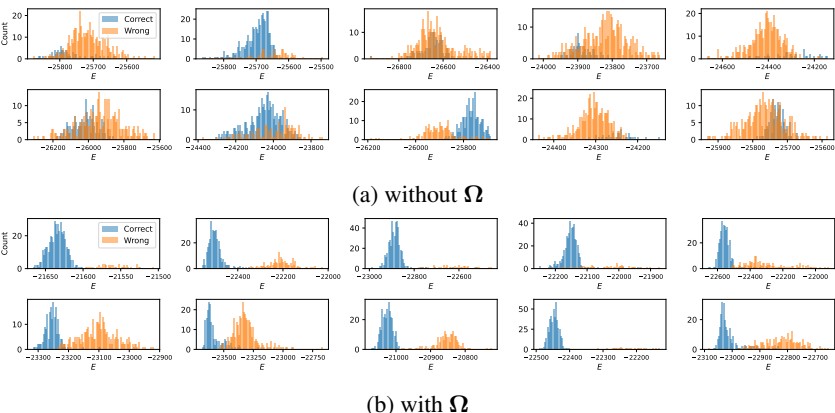

(a) without $\Omega$

(b) with $\Omega$

Figure 28: Energy distribution of the K-Net with or without the $\Omega$ term. In each panel, given a single board, we compute energies of the final oscillatory states that start from different random oscillators and show the histogram of these energies, color-coded by the correctness of the predictions made on the corresponding final oscillatory states.

Interestingly, the model without the $\Omega$ term does not give improvement with the energy vote, as the energy value and correctness are inconsistent (Fig. 28). This implies the asymmetric term $\Omega$ prevents the oscillators from being stuck in bad minima. We show the *AKOrN*'s board accuracy without the $\Omega$ term in Tab. 19, and see that the model's performance degrades with the energy vote (49.6 to 46.8).

### D.4 SYMMETRIC CONSTRAINT

Fig. 29 shows a comparison of *AKOrN* to Energy Transformer (Hoover et al., 2023) and a symmetric version of *AKOrN*. The symmetric version *AKOrN* is constructed by using the same weight to compute query and key vectors and a symmetric weight for value vectors. Board accuracies of these symmetrized models are shown in Tab. 19. We observe a similar tendency in the two symmetric models: both models underfit the data (See Fig. 29). Energy Transformer is not able to solve even in-distribution boards. Symmetrized *AKOrN* also gets stuck depending on the seed for the weight initialization.

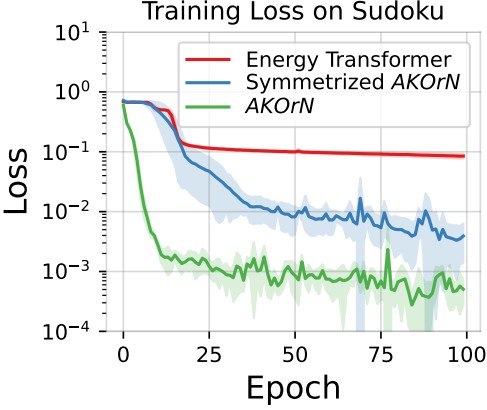

Figure 29: A training curve comparison with symmetric transformer models.

## D.5 ROBUSTNESS AND CALIBRATION ON CIFAR10

| Model | ↑ Accuracy | | | ↓ ECE |
|---|---|---|---|---|
| | Clean | Adv | CC | CC |
| Gowal et al. (2020) | 85.29 | 57.14 | 69.1 | 13.2 |
| Gowal et al. (2021) | 88.74 | 66.10 | 70.7 | 5.6 |
| Bartoldson et al. (2024) | 93.68 | 73.71 | 75.9 | 20.5 |
| Kireev et al. (2022) | 94.75 | 0.00 | 83.9 | 9.0 |
| Diffenderfer et al. (2021) | 96.56 | 0.00 | 89.2 | 4.8 |
| ViT | 91.44 | 0.00 | 81.0 | 9.6 |
| ResNet-18 | 94.41 | 0.00 | 81.5 | 8.9 |
| ItrConv | 93.46 | 0.00 | 83.6 | 5.9 |
| *AKOrN*$^{\mathrm{conv}}$ ($N = 2$) | 88.91 | *58.91 | 83.0 | 1.3 |
| *AKOrN*$^{\mathrm{mix}}$ ($N = 2$) | 91.23 | *51.56 | 86.4 | 1.4 |
| *AKOrN*$^{\mathrm{mix}}$ ($N = 4$) | 93.51 | *0.00 | 84.0 | 6.4 |

Table 20: (The extended table of Tab. 4) Robustness to adversarial attack (Adv) and Common Corruptions (CC) on CIFAR10 with the most severe corruption level (5). *The adversarial attack is done by AutoAttack with EoT (Athalye et al., 2018). The max norm constraint of the adversrial perturbtions is set to 8/255. With $N = 4$, the performance tendency of *AKOrN* is almost the same as ResNet except for the accuracy and uncertainty calibration on CIFAR10 with natural corruptions, which are moderately better with *AKOrN*$^{\mathrm{mix}}$.

d

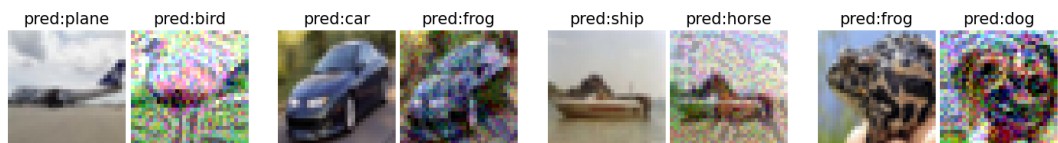

Figure 30: *AKOrN*'s adversarial examples are interpretable. Each pair of images is an original and the adversarially perturbed image ($\|\epsilon\|_\infty = 64/255$). The text above each image indicates the class prediction made by the *AKOrN* model.

# E    ADDITIONAL CLUSTER VISUALIZATIONS

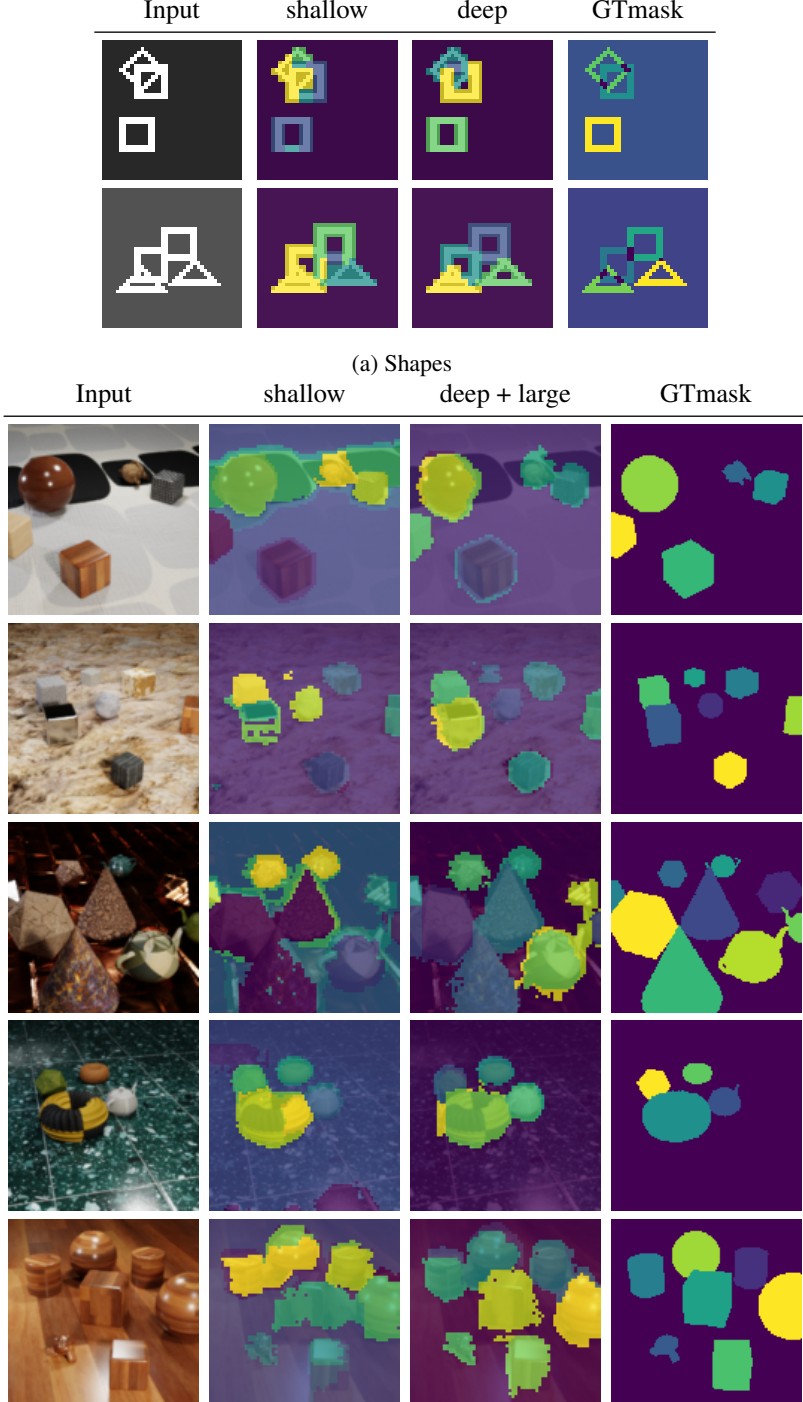

(a) Shapes

(b) CLEVRTex (1st and 2nd row), CLEVRTex-OOD (3rd and 4th row), and CLEVRTex-CAMO (the last row)

Figure 31: Deeper, wider, and more epochs make the models learn more binding features in *AKOrN*. (a): comparing a single-layer model (shallow) and a 3-layer model (deep). (b): comparing a single-layer model (shallow) and a model with doubled layers, channels, and epochs (deep+large).

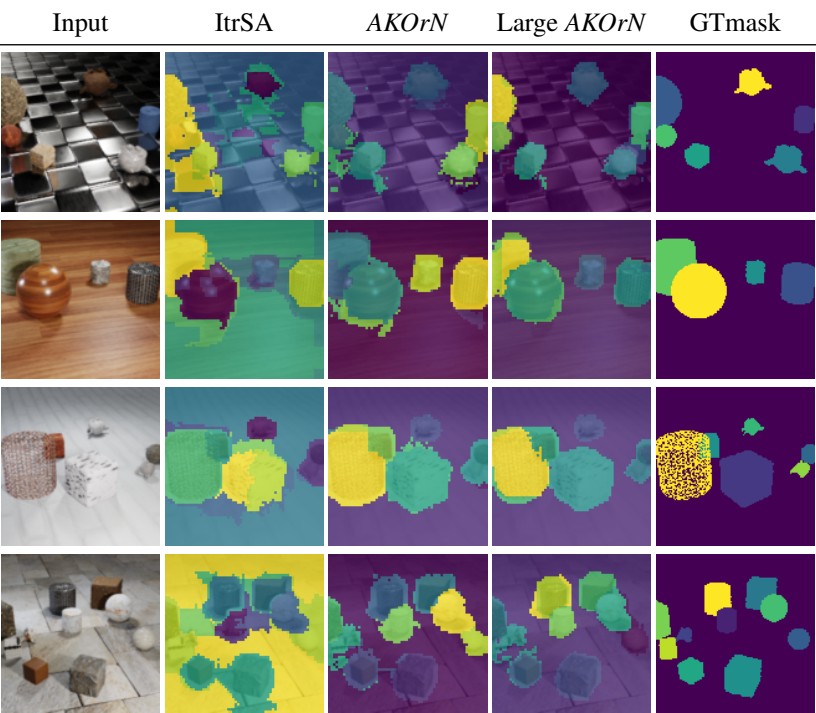

Figure 32: Visualization of clusters on CLEVRTex. The number of blocks $L$ is set to two for all models.

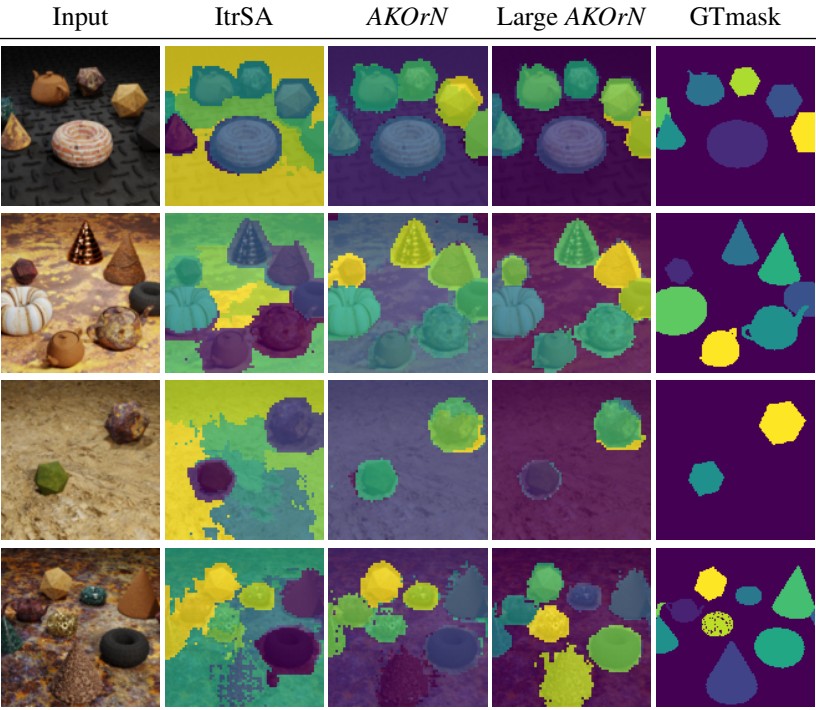

Figure 33: Visualization of clusters on CLEVRTex-OOD. The number of blocks $L$ is set to two for all models.

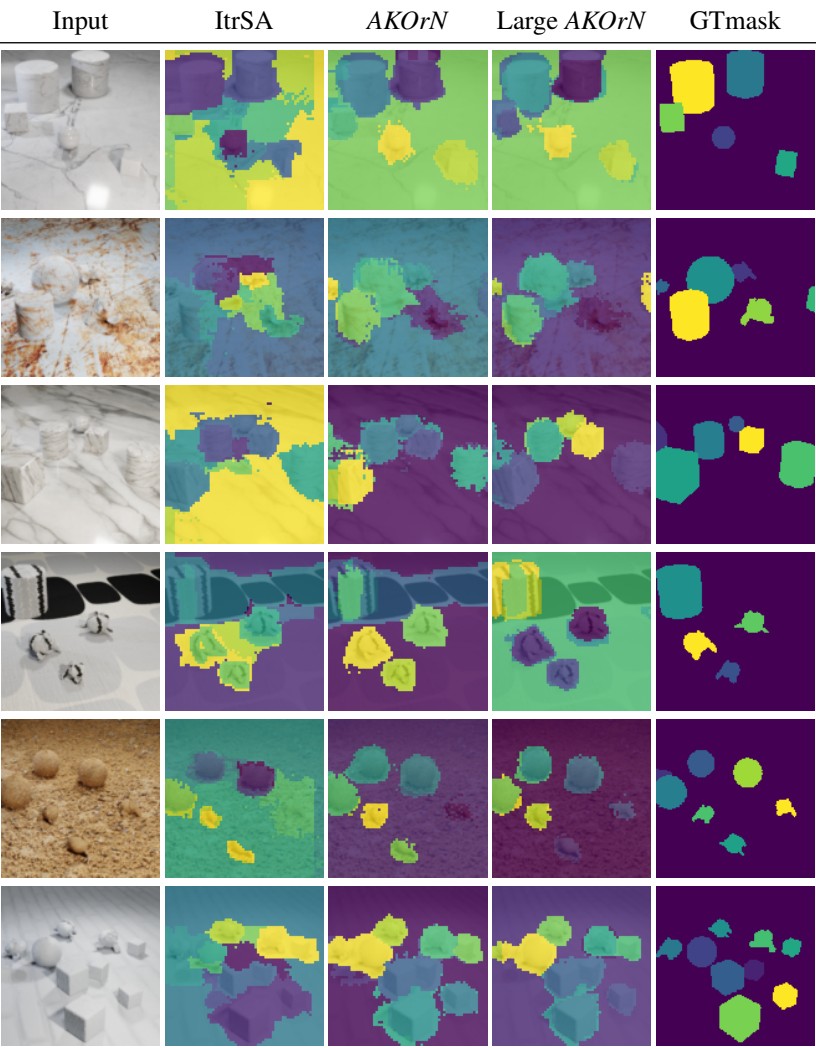

Figure 34: Visualization of clusters on CLEVRTex-CAMO. The number of blocks $L$ is set to two for all models.

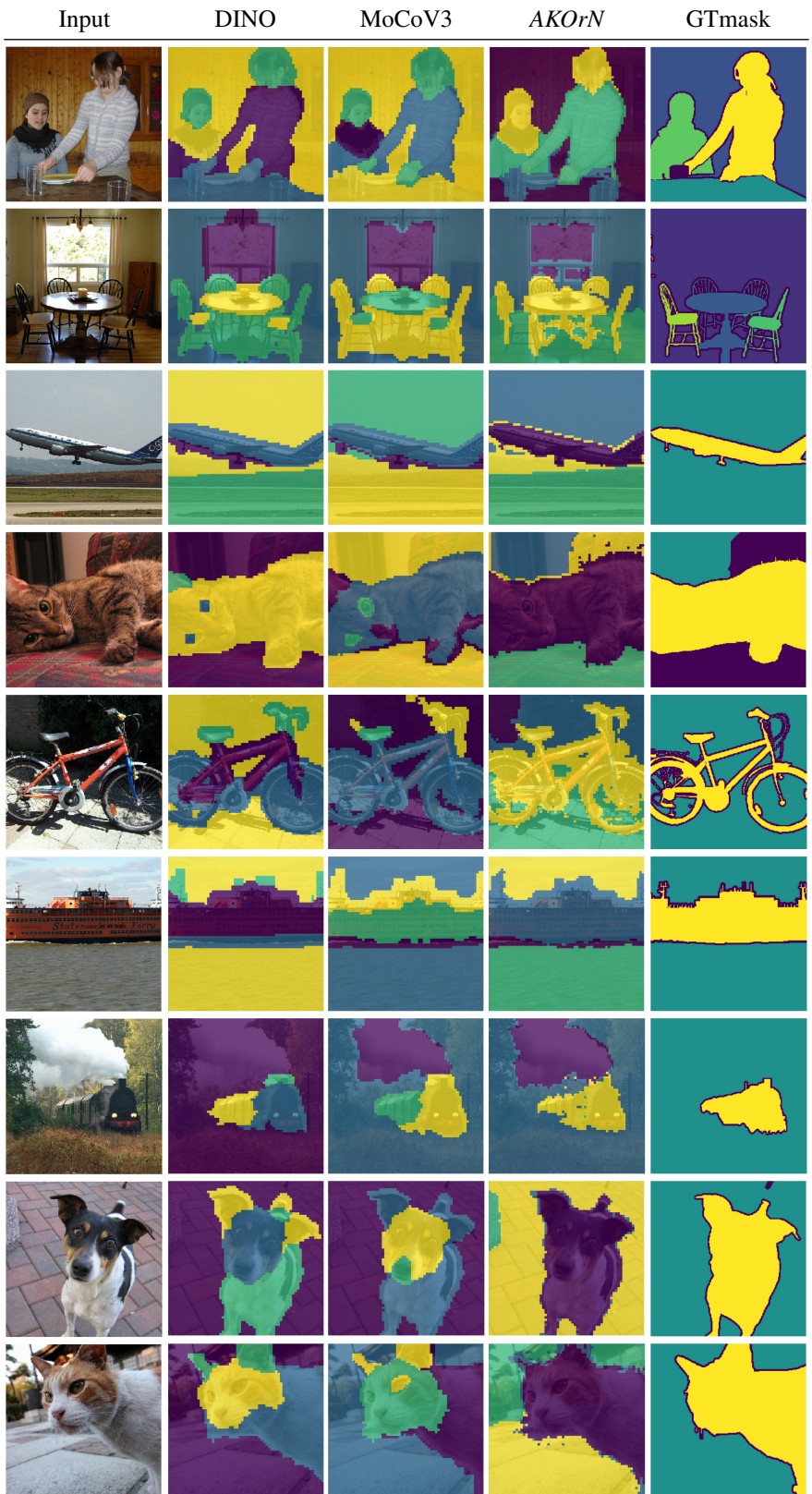

Figure 35: Visualization of clusters on PascalVOC. The number of clusters is set to 4.

| Input | DINO | MoCoV3 | *AKOrN* | GTmask |
|-------|------|--------|---------|--------|

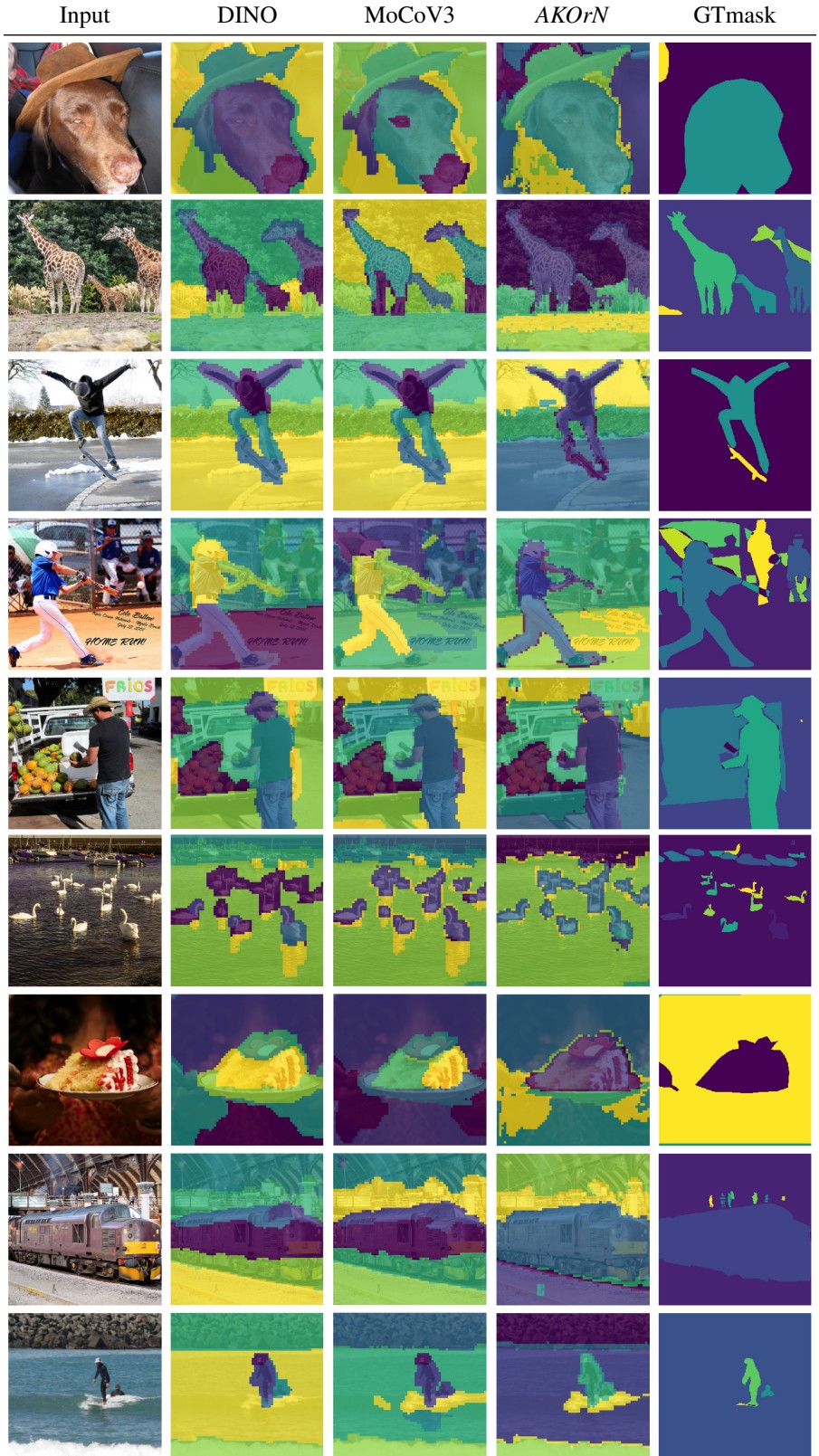

Figure 36: Visualization of clusters on COCO2017. The number of clusters is set to 7.

# F PROOF OF THE LYAPUNOV PROPERTY OF OUR GENERALIZED KURAMOTO MODEL

Under the assumptions: $\mathbf{J}_{ij} = J_{ij}\mathbf{I}$, $J_{ij} = J_{ji} \in \mathbb{R}$, $\mathbf{\Omega}_i = \mathbf{\Omega}$, and $\mathbf{\Omega}\mathbf{c}_i = \mathbf{0}$, we will prove below that Eq (3) is a Lyapunov function for the dynamics defined in Eq (2).

First, we compute the time derivative of $E$:

$$\frac{dE}{dt} = -\sum_i \left\langle \frac{\partial E}{\partial \mathbf{x}_i}, \dot{\mathbf{x}}_i \right\rangle = \sum_i \left\langle \sum_j J_{ij}\mathbf{x}_j + \mathbf{c}_i, \dot{\mathbf{x}}_i \right\rangle.$$

By substituting $\dot{\mathbf{x}}_i$ from the model,

$$\frac{dE}{dt} = -\sum_i \left\langle \sum_j J_{ij}\mathbf{x}_j + \mathbf{c}_i, \mathbf{\Omega}\mathbf{x}_i + \text{Proj}_{\mathbf{x}_i}\left(\sum_j J_{ij}\mathbf{x}_j + \mathbf{c}_i\right) \right\rangle,$$

which splits into two sums:

$$\frac{dE}{dt} = \underbrace{-\sum_i \left\langle \sum_j J_{ij}\,\mathbf{x}_j + \mathbf{c}_i, \mathbf{\Omega}\mathbf{x}_i \right\rangle}_{\text{(I)}} \underbrace{-\sum_i \left\langle \sum_j J_{ij}\,\mathbf{x}_j + \mathbf{c}_i, \text{Proj}_{\mathbf{x}_i}\left(\sum j J_{ij}\,\mathbf{x}_j + \mathbf{c}_i\right) \right\rangle}_{\text{(II)}}.$$

For the term (I), consider

$$\sum_i \left\langle \sum_j J_{ij}\,\mathbf{x}_j + \mathbf{c}_i, \mathbf{\Omega}\mathbf{x}_i \right\rangle.$$

We separate the $\mathbf{c}_i$ part:

$$\sum_i \langle \mathbf{c}_i, \mathbf{\Omega}\mathbf{x}_i \rangle = \sum_i \mathbf{c}_i^{\mathrm{T}}\mathbf{\Omega}\mathbf{x}_i.$$

Since $\mathbf{\Omega}\mathbf{c}_i = \mathbf{0}$ by assumption, it follows that $\mathbf{c}_i^{\mathrm{T}}\mathbf{\Omega} = 0$. Hence those terms vanish.

Next, for the $\sum_j J_{ij}\mathbf{x}_j$ part:

$$\sum_i \sum_j J_{ij}\,\mathbf{x}_i^{\mathrm{T}}\mathbf{\Omega}\,\mathbf{x}_j.$$

Since $J_{ij} = J_{ji}$ is symmetric but $\mathbf{\Omega}$ is skew-symmetric (i.e. $\mathbf{\Omega}^{\mathrm{T}} = -\mathbf{\Omega}$), one has

$$\mathbf{x}_i^{\mathrm{T}}\mathbf{\Omega}\,\mathbf{x}_j = -\mathbf{x}_j^{\mathrm{T}}\mathbf{\Omega}\,\mathbf{x}_i, \quad \Longrightarrow \quad J_{ij}\,\mathbf{x}_i^{\mathrm{T}}\mathbf{\Omega}\,\mathbf{x}_j = -J_{ij}\,\mathbf{x}_j^{\mathrm{T}}\mathbf{\Omega}\,\mathbf{x}_i.$$

Thus the double sum cancels term by term:

$$\sum_i \sum_j J_{ij}\,\mathbf{x}_i^{\mathrm{T}}\mathbf{\Omega}\,\mathbf{x}_j = 0.$$

Therefore, The term (I) is 0.

We are left with

$$\text{Term (II)} = -\sum_i \left\langle \sum_j J_{ij}\,\mathbf{x}_j + \mathbf{c}_i, \text{Proj}_{\mathbf{x}_i}\left(\sum_j J_{ij}\,\mathbf{x}_j + \mathbf{c}_i\right) \right\rangle.$$

Recall that $\text{Proj}_{\mathbf{x}_i}(\mathbf{y})$ is (by definition) the projection of $\mathbf{y}$ onto the subspace orthogonal to $\mathbf{x}_i$. In particular, if $\|\mathbf{x}_i\| = 1$, one has

$$\mathbf{y}^{\mathrm{T}}\text{Proj}_i(\mathbf{y}) = \left\|\text{Proj}_{\mathbf{x}_i}(\mathbf{y})\right\|^2 \geq 0$$

for all $\mathbf{y}$. Hence each inner product $\langle \mathbf{y}, \text{Proj}_{\mathbf{x}_i}(\mathbf{y}) \rangle \geq 0$.

Since there is a minus sign in front of this term in the expression for $\frac{dE}{dt}$ (see the original equation), we conclude Term (II) $\leq 0$.

Putting Term (I) and Term (II) together,

$$\frac{dE}{dt} = \underbrace{\text{Term (I)}}_{=0} + \underbrace{\text{Term (II)}}_{\leq 0} \leq 0.$$

Therefore, $\frac{dE}{dt} \leq 0$, proving that E is indeed a Lyapunov function for the given dynamics.

## G   MORE GENERAL PROOF

One can obtain natural sufficient conditions for energy function that ensure it is non-increasing on trajectory.

We define the block matrices $\boldsymbol{J}$ and $\Omega$ as follows:

$$
\boldsymbol{J} = \begin{bmatrix} \mathbf{J}_{11} & \mathbf{J}_{12} & \cdots & \mathbf{J}_{1C} \\ \mathbf{J}_{21} & \mathbf{J}_{22} & \cdots & \mathbf{J}_{2C} \\ \vdots & \vdots & \ddots & \vdots \\ \mathbf{J}_{C1} & \mathbf{J}_{C2} & \cdots & \mathbf{J}_{CC} \end{bmatrix}, \quad \Omega = \begin{bmatrix} \boldsymbol{\Omega}_1 & \cdots & \mathbf{0}_{N\times N} \\ \vdots & \ddots & \vdots \\ \mathbf{0}_{N\times N} & \cdots & \boldsymbol{\Omega}_C \end{bmatrix}.
$$

We similarly define $\boldsymbol{c}$ and $\boldsymbol{x}$.

**Proposition G.1** (sufficient conditions, non-constructive). *Suppose that the following conditions hold*

1. *Block matrix $\boldsymbol{J}$ is symmetric, i.e., $\boldsymbol{J}_{ii} = \boldsymbol{J}_{ii}^\top$, $\boldsymbol{J}_{ij} = \boldsymbol{J}_{ji}^\top$;*

2. *Block matrix $\Omega$ is block-diagonal and antisymmetric;*

3. *Block vector $\boldsymbol{c}$ is in the kernel of block matrix $\Omega$, i.e., $\Omega_i \boldsymbol{c}_i = 0$.*

*Energy function* (3) *is non-increasing on trajectories of dynamical system* (2) *if*

$$
\boldsymbol{J}\Omega - \Omega\boldsymbol{J} \geq 0, \tag{12}
$$

*that is commutator $[\boldsymbol{J}, \Omega]$ is a positive semi-definite matrix.*

*Proof.* Let $\boldsymbol{P}$ is a block-diagonal matrix with elements $\boldsymbol{P}_{ii} = \boldsymbol{I}_i - \boldsymbol{x}_i \boldsymbol{x}_i^\top$. Using block matrices we can rewrite dynamical system (2) and energy function (3) as $\dot{\boldsymbol{x}} = \Omega \boldsymbol{x} + \boldsymbol{P}(\boldsymbol{c} + \boldsymbol{J}\boldsymbol{x})$ and $E(\boldsymbol{x}) = -\frac{1}{2}\boldsymbol{x}^\top \boldsymbol{J}\boldsymbol{x} - \boldsymbol{c}^\top \boldsymbol{x}$. Using this notation we obtain for the derivative of the energy

$$
\frac{dE(\boldsymbol{x})}{dt} = \left(\frac{\partial E}{\partial \boldsymbol{x}}\right)^\top \dot{\boldsymbol{x}} = -\boldsymbol{x}^\top \boldsymbol{J}\dot{\boldsymbol{x}} - \boldsymbol{c}^\top \dot{\boldsymbol{x}} = -\boldsymbol{x}^\top \boldsymbol{J}\Omega\boldsymbol{x} - \boldsymbol{x}^\top \boldsymbol{J}\boldsymbol{P}(\boldsymbol{c} + \boldsymbol{J}\boldsymbol{x}) - \boldsymbol{c}^\top \Omega\boldsymbol{x} - \boldsymbol{c}^\top \boldsymbol{P}(\boldsymbol{c} + \boldsymbol{J}\boldsymbol{x}).
$$

The red term is zero by assumption. Blue terms can be combined together to the expression $\boldsymbol{y}^\top \boldsymbol{P}\boldsymbol{y}$ where $\boldsymbol{y} = (\boldsymbol{c} + \boldsymbol{J}\boldsymbol{x})$. This expression is always non-negative given that $\boldsymbol{P}$ is an orthogonal projector.

The remaining term $\boldsymbol{x}^\top \boldsymbol{J}\Omega\boldsymbol{x}$ can be transformed as $\frac{1}{2}\boldsymbol{x}^\top \left(\boldsymbol{J}\Omega + (\boldsymbol{J}\Omega)^\top\right)\boldsymbol{x}$. Using $\Omega^\top = -\Omega$ and $\boldsymbol{J} = \boldsymbol{J}^\top$ we obtain $\boldsymbol{x}^\top \boldsymbol{J}\Omega\boldsymbol{x} = \frac{1}{2}\boldsymbol{x}^\top [\boldsymbol{J}, \Omega]\boldsymbol{x}$. If this commutator is a positive-definite matrix we can be sure that the energy is non-increasing. □

The condition above is not especially convenient since it does not tells directly which matrices $\boldsymbol{J}$ and $\Omega$ are allowed. An example of a more direct result is given below.

**Proposition G.2** (sufficient conditions, constructive). *Commutator $[\boldsymbol{J}, \Omega] = 0$ and energy function* (3) *is non-increasing on trajectories of dynamical system* (2) *if $\boldsymbol{J} + \Omega$ is a normal matrix. One specific example be $\Omega = \boldsymbol{I} \otimes \Omega$ and $\boldsymbol{J} = \boldsymbol{k} \otimes \boldsymbol{I}$.*

*Proof.* We know that $\boldsymbol{J} + \Omega$ is a normal matrix. Using a standard definition of normality we obtain

$$
0 = (\boldsymbol{J} + \Omega)(\boldsymbol{J} - \Omega) - (\boldsymbol{J} - \Omega)(\boldsymbol{J} + \Omega) = -2[\boldsymbol{J}, \Omega].
$$

For a special choice $\Omega = \boldsymbol{I} \otimes \Omega$ and $\boldsymbol{J} = \boldsymbol{k} \otimes \boldsymbol{I}$ matrices $\Omega$ and $\boldsymbol{J}$ clearly commute since they are non-trivial in different factors of Kronecker product. □

The special case from Proposition G.2 is worth writing explicitly in the original notation. All oscillators have the same natural frequencies $\Omega_i = \boldsymbol{\Omega}$ and coupling matrices $\boldsymbol{J}$ realise multiplication by scalar, i.e., $\boldsymbol{J}_{ij} = k_{ij}\boldsymbol{I}$.

