# OpenReview forum: "Artificial Kuramoto Oscillatory Neurons"
_ICLR.cc/2025/Conference — ICLR 2025 Oral_

### Official Review · Reviewer_DAWQ · 2024-11-01

**Soundness:** 4
**Presentation:** 4
**Contribution:** 4
**Rating:** 10
**Confidence:** 4

**Summary:**

The paper proposes a novel type of neural network where neurons’ activities are defined on the unit sphere (in N-dimensional space, where N is typically 2 or 4), and are updated based on Kuramoto equations. This can model oscillatory dynamics that can be leveraged for binding activations across neurons. The model is applied to various settings (unsupervised object discovery, image classification adversarial or natural noise robustness, Sudoku problem solving) and compared against relevant baselines (e.g. slot-based models, Transformers), each time demonstrating competitive performance. The model has clear potential to improve the state-of-the-art across domains.

**Strengths:**

* A biologically inspired architecture, well described and motivated by neuroscientific findings
* A thorough set of experiments across diverse machine learning domains
* Meaningful comparisons with baselines revealing competitive performance

**Weaknesses:**

* The dependence of the model on the choice of dimensionality N, although it is explicitly acknowledged in the paper, warrants further investigations (which can be the goal of future studies)
* Although baselines are roughly matched for memory/parameter size and flops, it is possible that GPU optimizations will make them more efficient than the AKOrN versions. There is unfortunately no discussion of this issue nor any report of absolute time required for training or inference. This could be added to the Appendix.

**Questions:**

* on Lines 140-141, you argue that “we found that the use of C is necessary for stable trainings […] as a symmetry-breaking field”. But isn’t C also –and more simply—required for the network to receive inputs, and thus to perform any useful function?

---

> ### Author Response · Authors · 2024-11-21
> **Response to reviewer DAWQ**
>
> Thank you so much for rating our manuscript with such a high score, and thank you for the suggestion regarding the runtime comparison! For details on this, please refer to the General comments. In summary, runtime overhead in AKOrN can be negligible as the model size increases. We show the runtime comparisons on different datasets in Figure 17.
>
> > ***on Lines 140-141, you argue that “we found that the use of C is necessary for stable trainings […] as a symmetry-breaking field”. But isn’t C also –and more simply—required for the network to receive inputs, and thus to perform any useful function?***
>
> Thank you for pointing that out, and we apologize for not providing a full explanation. We noticed that referring to training-related aspects at this point was confusing, so we removed the sentence mentioning training stability from that paragraph.
>
> What we intended to convey is that the training became unstable when we removed $C$ and instead processed the input directly into $X^{(0)}$. This model is equivalent to just stacking Kuramoto layers and its performance is significantly worse than AKOrN (Please see Figure 16 in Section A.4 in the Appendix for details).

---

### Official Review · Reviewer_AtLx · 2024-11-02

**Soundness:** 3
**Presentation:** 3
**Contribution:** 3
**Rating:** 10
**Confidence:** 4

**Summary:**

- This paper developed a feature-rotation-like neuron models (high-dim feature rotation encodes synchrony binding), a promissingly neuron-level building block that can be combined with diverse network architectures, like CNN and transformer.
- The capability of the new architecture is validated on three independent tasks: object discovery (synthetic & natural image), reasoning on Sudoku task, robustness w.r.t adversarial attack in CI
FAR10.
- On each task, the proposed model shows SOTA performance and interesting shining points: (1) For object discovery, the model scales to natural image without pretrained DINO model; (2) for reasoning, the model's correctness is naturally implied by the "internal coherence", indicated by the task-independent energy measure; (3) for adversarial attack, robustness seems a built-in property of the dynamics and again, the energy could imply the accuracy.

**Strengths:**

I enjoyed reading this paper, impressed by the achievements on diverse tasks and also the authors's thoughts dispersed over the paper.  In brief, the strength of the paper includes:
- Soft grouping/clustering in neural network is an important question and often missing in main-stream ANNs.
- The proposed neuron model is neuroscientific motivated yet has concise fomula. The physical origin of the model makes it plausible to construct an "energy function", which turns out to be quite informative on the performance.
- While previous work on binding by synchrony mainly focuses on object discovery, the author extends these insights into reasoning tasks, and adversarial attack. And the performance on these new tasks is very successful.
- Lastly, the author provides a different conceptual angle for the binding by synchronization: the synchronization enables competitive learning, and therefore compresses representation to construct a bottle neck, which facilitate abstraction. As far as I know, it is a novel argument not presented in previous works and it also makes sense to me.

**Weaknesses:**

For the weakness:
- It is not clear what is the actual conceptual contribution / nolvelty of the Kura model proposed in this paper compared with Lowe's Rotating Feature (and recent updates). For example, equation (6) is very similiar to the "binding mechanism activation" in  [1] and I have the intuition that this is the essential mechanism for the binding ability of model, similiar to [1]. If I am wrong, please correct me.
- When generalizing the original Kura model to high-dim, it is not clear whether the Proj in eq(2) is an 'equivalent' generalization of $sin(\theta_j-\theta_i)$. So it is not clear how much this term actually contribute to the computation of the network. Note that $sin(\theta_j-\theta_i)$ is a essential mechanism in original Kura model to guarentee the synchrony as a stable state and sin acts among each pair of neurons before the $\sum$, but Proj only acts after the $\sum$. (The first two concerns are mainly about how "Kura" the model is ?)
- Since the activity is constraints on the sphere, it loses the representational ability for encoding feature presence, which is an important conceptual diaviation from synchrony binding idea and is likely to limit its ability to infer in certain cases, e.g. when the feature is uncertain or weakly related to the task.
- On the experimental side, it mostly ignored the Lowe's model for comparison (Fig.3, Fig.4,Fig.5,Tab.1,Tab2, Tab3...) and other synchrony based model, e.g. [2].
- The code is not provided, so it is hard to evaluate the the reproducibility of the results.
- The author misses several essential recent works: e.g. [2] shows how oscillation emerges in spike-based model to bind features; [3] combine Kura model with ANNs and also achieved synchrony binding.
- The first two sentences in Abstract is quite confusing and the first paragraph in Introduction is not very informative.

[1] Löwe S, Lippe P, Locatello F, et al. Rotating features for object discovery[J]. Advances in Neural Information Processing Systems, 2024, 36.
[2] Zheng H, Lin H, Zhao R. GUST: combinatorial generalization by unsupervised grouping with neuronal coherence[J]. Advances in Neural Information Processing Systems, 2024, 36.
[3] Ricci M, Jung M, Zhang Y, et al. KuraNet: systems of coupled oscillators that learn to synchronize[J]. arXiv preprint arXiv:2105.02838, 2021.

**Questions:**

(1) For the claim in the paper:
> “binding” between neurons leads to a form of competitive learning where representations are compressed  in order to represent more abstract concepts in deeper layers of the network.

> In fact, neighboring neurons tend to cluster their activities, and clusters tend to compete to explain the input. This “competitive learning” has the advantage that information is compressed as we move through the layers, facilitating the process of abstraction by creating an information bottleneck. Additionally, the competition encourages different higher-level neurons to focus on different aspects of the input (i.e., they specialize). This process is made possible by synchronization: like fireflies in the night, neurons tend to synchronize their activities with their neighbors’, which leads to the compression of their representations.

Can author clarify the reference if such claim is well-documented? If it is the novel claim of the author, the author should modify the tone to avoid overclaim or provide rationale or experiments to make sense of such claims.

(2) (Weakness section). What is the contribution of the "Proj", and will there be synchronization, e.g. given input image, if we replace Proj into general linear sum (ablation of Proj)? And what is the difference between eq(6) and binding mechanism in Lowe's work? Since the binding mechanism itself can generate synchronization, how to disentangle the contribution from (6) to contribution from the "Kura idea".

(3) Can author shows the segmentation results on binary datasets, like Zheng2024. The different color of object may make it eaiser to break symmetry compared with binary images. e.g. in Fig.4 the segmentation is more likely based on color instead of object-instance (the highlight is treated as an object). Besides, what is the actual input dimmension of the image to the model? To what extent it is downscaled?
(4) Why the energy function still make sense even if the J and $\Omega$ is not constraint by symmetry?
(5) What is the rationale of robustness? Is it the general outcome of dynamical models or is it the outcome of this Kura model? I suggest compare with related non-synchrony dynamical ANN models to disentagle the causal-effect.
(6) Why energy function, or the synchrony level in the network, can indicate the task performance? Can the author explain that more solidly?
(7) For the Sudoku task, can the author compare with those Ising-based models? e.g. Hopfield network or modern hopfield network. So as to disentagle the contribution for the "Kura idea" from that generally comes from attractor dynamics.

In general, I would like to vote for the accept of the paper, but I will also revisit the evaluation based on the author's response and other reviewer's comments.

---

> ### Author Response · Authors · 2024-11-21
> **Response to reviewer AtLx**
>
> We are glad that you enjoyed reading the manuscript! We will answer the concerns and questions you raised one by one.
>
> > ***It is not clear what is the actual conceptual contribution / novelty of the Kura model proposed in this paper compared with Lowe's Rotating Feature (and recent updates). For example, equation (6) is very similiar to the "binding mechanism activation" in [1] and I have the intuition that this is the essential mechanism for the binding ability of model, similiar to [1]. If I am wrong, please correct me.***
>
> Let us first clarify that the Lowe's rotating feature (RF)’s activation is very different from just "taking norm" used in our Readout module.  The activation used in RF is the sum of the two following terms: "the norm of the weighted sum of rotating neurons" and "the weighted sum of the norm of neurons". The latter term is essential in RF and called xi-term [i][ii], which enables the model to learn binding features. [ii] reported no binding is observed without xi-term (See the “-χ” of Table 3 in [ii] which shows the ablation of the xi-term). Our norm term in the Readout module is more similar to the former one: "the norm of the weighted sum of rotating neurons".
>
> The norm term in our Readout module is used mainly for the purpose of stabilization of training. When AKOrN is trained on CIFAR10 without the norm term, it significantly underfits the data, resulting in a 7% drop in accuracy compared to the full AKOrN model (See Figure16 and Section A.4).
>
> As an ablation study, we test a model replacing the Kuramoto mechanism with the conventional residual update. We see the model degrades significantly in both the object discovery performance and Sudoku solving, as shown in the table below, which clearly shows the large Kuramoto model’s contribution to the performances. The table is included in the updated manuscript as Table 6.
>
> ### CLEVR-Tex
> | Kuramoto | FG-ARI | MBO  |
> |----------|--------|------|
> | ✗        | 65.0   | 51.3 |
> | ✓        | 81.5   | 54.1 |
>
> ### Sudoku
> | Kuramoto | ID        | OOD         |
> |----------|-----------|-------------|
> | ✗        | 59.8±54.6 | 17.1±16.6   |
> | ✓        | 100.0±0.0 | 51.7±3.3    |
>
> At a high level, Rotating Features do a simple forward pass through the model, where the orientations of the features are set statically via the rotation bias, and feature binding is through a specialized activation mechanism called xi-binding. In the Kuramoto model, we explicitly model the temporal component by implementing an iterative procedure within each layer that promotes synchronization.
>
> ---
> > ***When generalizing the original Kura model to high-dim, it is not clear whether the Proj in eq(2) is an 'equivalent' generalization of sin(θj−θi)***.
>
> We show the relation between the vectorized Kuramoto model and the original one in Appendix A.1 along with a visualization of the projection operator. In summary, the vectorized Kuramoto model includes the original one in a special case where $N=2$,  $c_i=0$, and the connectivity $J_{ij}$ is a scalar multiple of the identity matrix.
>
> > ***sin(θj−θi)  is a essential mechanism in original Kura model to guarentee the synchrony as a stable state and sin acts among each pair of neurons before the ∑, but Proj only acts after the ∑.***
>
> The order of the application of ∑ and Proj does not matter because Proj is a linear operator (${\rm Proj}_x=I - xx^{\rm T}$) and thus Proj ∑ = ∑ Proj. The same is applied to the original Kuramoto model: $\sum_j {\rm sin}(\theta_j−\theta_i) = Im [\sum_j e^{\sqrt{-1}(\theta_j−\theta_i)}] =  Im [e^{-\sqrt{-1}\theta_i} \sum_j e^{\sqrt{-1}\theta_j} ]$

---

> ### Author Response · Authors · 2024-11-21
> **Response to reviewer AtLx (cont.)**
>
> > ***What is the contribution of the "Proj", and will there be synchronization, e.g. given input image, if we replace Proj into general linear sum (ablation of Proj)?.***
>
> Thank you for your suggestion! We use Proj to align the vectorized version with the original Kuramoto model, as we describe in the above comment. Proj ensures the update direction stays on the tangent space of the sphere. Note that with or without Proj only changes the length of the actual update direction of each neuron. The updated $x_i$ stays on the sphere since we normalize the updated neuron to be the unit vector in Eq (5).
>
> We test AKOrN without Proj operators and summarize the results below. We see almost identical, and slightly degraded performance on the CLEVR-Tex object discovery experiment and on the Sudoku solving, respectively. Interestingly, without projection, the AKOrN’s adversarial robustness and uncertainty quantification get worse compared to the original AKOrN.
>
> ### (a) CLEVR-Tex object discovery
> | Proj$_x$ | FG-ARI | MBO  |
> |--------|--------|------|
> | ✗      | 80.9   | 57.4 |
> | ✓      | 81.5   | 54.1 |
>
> ### (b) Sudoku board accuracy
> | Proj$_x$ | ID        | OOD         |
> |--------|-----------|-------------|
> | ✗      | 99.9±0.0  | 45.0±1.9    |
> | ✓      | 100.0±0.0 | 51.7±3.3    |
>
> ### (c) CIFAR10 classification
> | Proj$_x$ | ↑ Acc  | |      | ↓ ECE |
> |--------|------|------|------|-------|
> |        | Clean | Adv | CC | CC    |
> | ✗      | 89.9  | 0.1 | 82.4 | 4.5 |
> | ✓      | 84.6  | 64.9 | 78.3 | 1.8 |
>
> > ***Since the activity is constraints on the sphere, it loses the representational ability for encoding feature presence***
>
> Thank you for your comment! We would like to note that although each single oscillator cannot encode the feature presence, a group of oscillators can learn such feature presence through the degree of their coherence: e.g. we can think of the magnitude of the sum of a set of oscillators as feature presence. This way to encode feature presence is possible through our readout module that takes the norm of the weighted sum of oscillators.
>
> > ***On the experimental side, it mostly ignored the Lowe's model for comparison (Fig.3, Fig.4,Fig.5,Tab.1,Tab2, Tab3...) and other synchrony based model***
>
> The Lowe's Rotating features [iii] and other synchrony-based models’ [ii, iv, v] results are shown in Tab. 14 in the appendix, and AKOrNs are consistently better than these models on all synthetic datasets, except for dSprites on which AKOrN and Rotating features are on par. We show the Rotating features (RF)’s results on PascalVOC on Table 18, and AKOrN is significantly better than RF.  Note that no synchrony-based models besides AKOrNs are directly applicable to natural images. Even on CLEVR data, which is one of the synthetic datasets, Lowe’s model and other synchrony-based models get considerably worse performance than slot attention and AKOrN (See also Figure16 in Lowe's work [iii] where the model separates the objects based on low-level features such as colors). The model in [2] the reviewer suggested is only tested on simplistic, synthetic data, and it would be unlikely to scale to complex datasets such as CLEVR, CLEVR-Tex, and natural images. However, if you would still like to see the results of [2] on these datasets, we are willing to train [2]’s model on our dataset and report the performance.
>
> > ***The code is not provided, so it is hard to evaluate the the reproducibility of the results.***
>
> We are currently preparing the reproducing code and will provide a part of the code for the object discovery experiments as soon as possible. We are committed to open-sourcing the entire codebase, including scripts for reproducing all the experiments conducted in our work, at a later date.
>
> > ***The author misses several essential recent works: e.g. [2] shows how oscillation emerges in spike-based model to bind features; [3] combine Kura model with ANNs and also achieved synchrony binding.***
>
> Thank you for the references! We apologize for having missed these works. We discuss them in the related work section of the updated manuscript.
>
> >***The first two sentences in Abstract is quite confusing and the first paragraph in Introduction is not very informative.***
>
> Thank you for your comments! Could you elaborate a bit more about why the abstract is confusing, and the first paragraph is not informative? We are a bit confused about this comment. The first two sentences in the abstract describe the core idea and motivation behind our proposed neuronal mechanism. The first paragraph of the introduction mentions the McCulloch-Pitts neuron mechanism which is the current main paradigm of deep neural networks but ignores the temporal dynamics of the real neurons. The paragraph explains the very motivation of our temporal processing neurons that is inspired by the recent findings in neuroscience.

---

> ### Author Response · Authors · 2024-11-21
> **Response to reviewer AtLx (cont.)**
>
> - ***Regarding references to the claim that we made in the first sentence of the abstract and the first paragraph of the motivation section***
>
> (This is our response to the reviewer's comment "Can author clarify the reference if such claim is well-documented?")
>
> Thank you for your suggestions! There are many, but let us name a few. We include these references in the motivation section in the updated manuscript.
> - Amari S. (1982). Competitive and cooperative aspects in dynamics of neural excitation and self-organization. In Competition and Cooperation in Neural Nets: Proceedings of the US-Japan Joint Seminar held at Kyoto, Japan February 15–19, 1982 (pp. 1-28). Berlin, Heidelberg: Springer Berlin Heidelberg.
> - Grossberg S., and Grunewald A. (1997). Cortical synchronization and perceptual framing. Journal of cognitive neuroscience, 9(1), 117-132.
> - Gray et al. (1989). Oscillatory responses in cat visual cortex exhibit inter-columnar synchronization which reflects global stimulus properties. Nature, 338(6213), 334-337.
>
> We also experimentally show results aligned with the claim: the deeper the layers, the more object-aligned binding features the model learns (See Tables 15,16 and Figure 29).
>
> > ***Can author shows the segmentation results on binary datasets, like Zheng2024. The different color of object may make it eaiser to break symmetry compared with binary images…***
>
> (Nov. 21, 11:34 GMT: We realized that we had forgotten to include the performance table and reference the figure showing the clustering results in this rebuttal comment. We have now added them. We apologize for overlooking these details.)
>
> Thank you for the suggestion! We manually created a binary image dataset, named “Shapes,” consisting of images with 2–4 objects that are randomly sampled from four basic shapes (triangle, square, circle, and diamond). Note that each image can have multiple objects of the same shape together. The table below shows that AKOrN is considerably better than its counterpart model (ItrSA). The same table is included in the updated manuscript as Table 15.
>
> #### (a) FG-ARI
> | \(L\)    | 1    | 2    | 3    |
> |----------|------|------|------|
> | ItrSA    | 47.5 | 48.3 | 49.2 |
> | AKOrN$^\text{attn}$ | **56.2** | **63.6** | **72.6** |
>
> #### (b) MBO
> | \(L\)    | 1    | 2    | 3    |
> |----------|------|------|------|
> | ItrSA    | 38.5 | 39.0 | 30.1 |
> | AKOrN$^\text{attn}$ | **41.6** | **45.0** | **48.8** |
>
> Figure 24 demonstrates that the AKOrN model learns significantly better object-binding features than ItrSA on this highly symmetric dataset. Furthermore, having more layers benefits the separability in AKOrN while doing little for ItrSA.
> The primary reason we do not achieve perfect results here would be the use of patchified images, which makes it difficult for the model to assign fine object details to coarse features accurately.
>
> > ***Fig.4 the segmentation is more likely based on color instead of object-instance (the highlight is treated as an object)***.
>
> We would like to emphasize that while the model is actually distracted by the strong highlight, the objects in Figure 4 of the initial submission are very similar in color, which highlights the opposite of the concern: the segmentation is unlikely to be based on the objects’ color. The results on highly symmetric datasets, as mentioned above, further support this point.
> Additionally, it has become evident that AKOrN has learned improved object-binding features thanks to the newly introduced up-tiling method. In the updated manuscript, Figure 4 on the right side shows the AKOrN is not misled by the shiny, object-like white highlight on the floor, even in a scene where both the objects and the floor are white.
>
> We observe that deeper and wider AKOrN models are less misled by such superficial features. Figure 29(b) shows that a larger AKOrN model is not deceived by an irregular floor pattern, a mirroring object, and a highlighted object (1st, 2nd, and 5th row, respectively).
>
> > ***Besides, what is the actual input dimension of the image to the model? To what extent it is downscaled?***
>
> Here is a table that summarizes the size of the images, the number of channels, and patch-size (= the downscale factor). The same information is shown in Tabs 9-11 in the Appendix.
>
> |                | Tetrominoes | dSprites | CLEVR | Shapes | CLEVRTex (-OOD, -CAMO) | ImageNet (Pascal, COCO) |
> |----------------|-------------|----------|-------|--------|------------------------|-------------------------|
> | Image resolution | 32         | 64       | 128   | 40     | 128                    | 256                     |
> | Patch size       | 4          | 4        | 8     | 2      | 8                      | 16                      |
> | Feature resolution | 8        | 16       | 16    | 20     | 16                     | 16                      |
> | Feature size      | 128        | 128      | 256   | 256    | 256/512                | 768                     |

---

> > ### Author Response · Authors · 2024-11-21
> > **Response to reviewer AtLx (cont.)**
> >
> > > ***What is the rationale of robustness? Is it the general outcome of dynamical models or is it the outcome of this Kura model? I suggest compare with related non-synchrony dynamical ANN models to disentagle the causal-effect***
> >
> > Thank you for your suggestion! We train a recurrent model that is similar to AKOrN but uses a conventional residual update instead of the Kuramoto update. This is almost identical to an ItrConv model used in the object discovery experiment (see the last paragraph in Section D.3 for details about the architecture). Table 19 in the Appendix shows that ItrConv has almost no adversarial robustness and worse uncertainty quantification than $\textit{AKOrN}^{\rm conv}$.
> >
> > Also, we conducted an analysis on the rationale of the robustness and observed highly fluctuating oscillatory states over timesteps in AKOrN. Please refer to the General comments for details.
> >
> > > ***Why the energy function still make sense even if the J and Ω is not constraint by symmetry?***
> > > ***Why energy function, or the synchrony level in the network, can indicate the task performance?***
> >
> > Thank you for raising these questions! Please see the General comments where we elaborate on why the energy value can tell the solution's correctness.
> >
> > > ***(7) For the Sudoku task, can the author compare with those Ising-based models? e.g. Hopfield network or modern hopfield network. So as to disentagle the contribution for the "Kura idea" from that generally comes from attractor dynamics.***
> >
> > As indicated in “the Hopfield is all you need” paper (vi), the modern Hopfield layer becomes self-attention (See Sec. 3 (1) on page 6 of “the Hopfield is all you need” when replacing the stored patterns with the input). This is exactly the ItrSA model we tested in our work, where we apply shared self-attention layers (=Hopfield layers) across timesteps. We show its inferior performance to AKOrN throughout the paper. One significant difference from the usual Ising models is the asymmetry. To see the efficacy of allowing asymmetry in AKOrN, we compare it with Energy transformer (ET) [vii] and a symmetrized AKOrN.  ET symmetrizes the self-attention so that the update ensures minimizing a certain energy (Eq.(3) in [vii]). Symmetrized AKOrN is the same as AKOrN but shares Q,K weights and has a symmetric value weight. Please kindly refer to Figure 27 showing the learning curves of ET, Symmetrized AKOrN, and the original AKOrN. We see that AKOrN minimizes the loss much faster than the symmetrized models. We also observed that one of 5 symmetrized AKOrN models with different random seeds gets stuck during training (we excluded that model from the graph).
> > Table 18 in the Appendix shows that board accuracies of the symmetrized models are much worse than AKOrN. ET gets almost zero board accuracy, even on in-distribution boards.
> >
> > ## References
> > - [i] Reichert D. and Serre T. Neuronal synchrony in complex-valued deep networks. arXiv preprint
> > arXiv:1312.6115, 2013
> > - [ii] Löwe et al. Complex-valued autoencoders for object discovery. arXiv preprint arXiv:2204.02075, 2022.
> > - [iii] Löwe et al. Rotating features for object discovery. In Advances in Neural Information Processing Systems (NeurIPS), 2023
> > - [iv] Stani´c et al. Contrastive training of complex-valued autoencoders for object discovery. In Advances in Neural Information Processing Systems (NeurIPS), 2023.
> > - [v] Gopalakrishnan et al. Recurrent complex-weighted autoencoders for unsupervised object discovery. arXiv.org, 2024.
> > - [vi] Ramsauer et al. Hopfield networks is all you need. arXiv.org, 2020.
> > - [vii] Hoover et al. Energy transformer. In Advances in Neural Information Processing Symtems (NeurIPS), 2023.

---

> ### Author Response · Authors · 2024-11-24
> **Reproducing code**
>
> Dear Reviewer AtLx,
>
> We here provide the code to reproduce the CLEVR-Tex results:
>
> [https://anonymous.4open.science/r/akorn-049D](https://anonymous.4open.science/r/akorn-049D)
>
> We hope this will help clarify and validate the results presented in our work. Please feel free to reach out if you have any further questions or require additional details.
>
> Best,
> Authors

---

> > ### Comment · Reviewer_AtLx · 2024-11-24
> > **Response to the author**
> >
> > Thank you very much for the explanation, which resolved my biggest concerns: the difference from previous works, the role of "Kura" component and the supports of arguments in the main text. Since scaling soft-binding mechanism itself is a big challenge and the mechanism proposed in this paper is very general, I hope to recommend this work as a highlight in the conference. Therefore, I raise my rating to 10.
> >
> > A minor question: I wonder why the performance on such a simple Shapes dataset is so low (0.56 to 0.72), even after the background is removed. Does it mean that binary image poses bigger challenge for the current model? I also wonder what is the score if the background is not removed during evaluation (figure-background problem).

---

> > > ### Author Response · Authors · 2024-11-25
> > >
> > > Thank you very much for your feedback and for significantly raising the score! We sincerely appreciate your acknowledgment of our work’s contribution to scaling synchrony-based models.
> > >
> > > >  why the performance on such a simple Shapes dataset is so low (0.56 to 0.72)
> > >
> > > We think this is due to the patchification of images into coarse representations. On such coarse representations, the model has to assign mixed object information on each fixed-size vector token, which is inherently challenging. Most failure cases for both AKOrN and non-Kuramoto-based models (ItrSA) occur with images having intricate object overlaps (See Figure 24(b)). When we compute the scores on images with completely separated objects, FG-ARI increases from 72 to 85 with AKOrN(L=3).
> > >
> > > > I also wonder what is the score if the background is not removed during evaluation (figure-background problem).
> > >
> > > Both AKOrN and ItrSA perform well in separating the background from objects. IoU between each background mask and the cluster that overlaps with it the most is ~90. MBO increases from 48 to 60 by including the background in evaluation.

---

### Official Review · Reviewer_ynsZ · 2024-11-02

**Soundness:** 3
**Presentation:** 2
**Contribution:** 3
**Rating:** 8
**Confidence:** 3

**Summary:**

the authors propose a neural ntwork architecture in which units are high-dimensional Kuramoto oscillatory neurons. This architecture leads synchronization or 'binding' between neurons which enables competitive performance in object discovery, reasoning and robustness.

**Strengths:**

- the model is interesting and well defined
- the quantitative results appear impressive, though I am have some issues with their presentation (see below)
- the many numerical results presented, and comparisons to other models, suggests that overall a good deal of thought/time has been put in the manuscript

**Weaknesses:**

- there are numerous minor issues with the text/presentation (see question below) which impacts the clarity of the paper
- This not a field I have experience in, but I am not entirley convinced a fair comparison was made to other models for object discovery. Is the idea that AKOrN is a highly competitive model for models 'trained from scratch', and that comparing it to models with some pretrained parameters (e.g. Lowe et al. 2024) is unfair? I think this could be better clarified if so
- there is a notion from the abstract and sections 1/2 that this work is relevant to neuroscience, but the link feels quite tenuous to me. Is it reasonable to think of neurons as high-d points on a sphere?

**Questions:**

- The format of figure references could be improved. I would recommend references as 'Fig. X'. Currently there is either the period missing or there's no space (e.g. 'Fig1' on line 96). Same with Tab X -> Table X. Should also be Figs for multiple figures (not Fig as in line 368)
- line 36: 'we follow a more modern dynamical view of neurons as oscillatory units'. Is this correct? Oscilliations have been studied in the brain for a long time; moreover, I would say that the majority of computational neuroscientists currently model neurons as scalar values.
- For table 3 results are presented with error bars over 5 seeds (which I favour), why is this not the case for any of the other results presented?
- line 121: 'we introduce a multi-dimensional vector version of the model' - this line is confusing as this was already employed by Chandra et al. and Lipton et al. (as cited). Perhaps replace with 'incorporate a mutli-dimensional model...'. On this subject could you clarify to me the exact novelty of the model? Is it that it's not been introduced in deep neural nets before? Or is it about the fact connections between oscillators J is not symetric in AKOrN? Relatedly, line 308 'our..models differ from these approaches in various aspects such as...' should really provide all fundemental differences beyond some examples. In fact, in any case what's the difference between 'asymmetric connections' and 'their symmetry breaking term'?
- line 123 (and other places): what's C? the size of the neural population? this should be clarified
- line 245: what is 'proper' energy?
- line 323: I don't understand how these convolution/self-attention layers are iterated upon. What are the shared parameters?
- I didn't understand the implementation for the sudoku task. 1. is the entire 9x9 soduku grid fed all at once, or one digit at a time? and 2. how come x_i takes the value c_i when a digit is given, does the model no longer follow equation 2? Or is this just the initial value taken before the Kuramoto steps?
- line 425: 'the energy is a good indication of the solution's correctness'. Why? I can believe that stability is often a desirable thing in tasks but think this line deserves some elaboration
- line 464: would it be possible to give a brief description/overview of what 'common corruptions' are?
- line 506: why is it surprising that confidence and accuracy are linearly correlated?
- line 527: 'the oscillator...'. I didn't understand the logic of this sentence, could you explain it to me please?

Other comments:

- the notation x_{chw} is confusing, I think x_{c,h,w} is better
- on line 201 the terms should be switched around (the initial state of the next block is the thing being defined). But also, how does this fit in with line 189 where X^(l,0) = X^l?
- the metrics in all of the tables should be clarified. I presume the unit is %, but e.g. in the case of Table 1 I'm not entirely sure what the metric is. % objects correctly identified?
- in the results section the language can be overly strong. For example 'far better' (line 377) and 'vastly better' (line 416) do not appear to me as very scientific language.
- in the discussion there is talk of small N resulting in a strongly regularized model. Given that the size of N nor regularization was mentioned in the prior text, this comes rather abruptly as is confusing to the reader

typos:
- line 150: 'set C a silhouette' -> 'set C as a silhouette'

---

> ### Author Response · Authors · 2024-11-21
> **Response to reviewer ynsZ**
>
> Thank you so much for your valuable comments and suggestions, which are greatly helpful in improving our paper quality. We address your questions and concerns below, one by one.
>
> > ***Is the idea that AKOrN is a highly competitive model for models 'trained from scratch', and that comparing it to models with some pretrained parameters (e.g. Lowe et al. 2024) is unfair?***
>
> Indeed, we believe the direct comparison to models that are applied to large-scale, pre-trained models is not fair. AKOrN achieves state-of-the-art object discovery results on real-world images while being trained from scratch. In contrast, to the best of our knowledge, none of the existing object discovery methods perform considerably better than random in this regime.
> Note that in Table 1 and Table 18 in the Appendix section, we show that AKOrN now is better than such models with pre-trained parameters including (Lowe et al. 2024), DINASOUR, Slot-diffusion, and SPOT on Pascal, and better on COCO than those methods except for SPOT.
>
> > ***there is a notion from the abstract and sections 1/2 that this work is relevant to neuroscience, but the link feels quite tenuous to me. Is it reasonable to think of neurons as high-d points on a sphere?***
>
> > ***line 36: 'we follow a more modern dynamical view of neurons as oscillatory units'. Is this correct? ... I would say that the majority of computational neuroscientists currently model neurons as scalar values.***
>
> People modeling oscillatory neurons often represent them by complex values (or equivalently by the sinusoidal function), especially when they investigate synchronization in neuron populations (e.g. [i]). The complex-valued neuron is quite close to our generalized Kuramoto oscillator with N=2. The N=2 model is equivalent under certain symmetric constraints, as we commented in the responses to reviewer AtLx. The AKOrN model with N=2 exhibits good adversarial robustness. When we extend the Kuramoto oscillators to  N>2, we see an improvement in the model capacity and flexibility and achieve good performance on object discovery and Sudoku solving. In the end, we take inspiration from neuroscience and physics, but we take the liberty to deviate from these inspirations whenever this can lead to improved results. For example, we also do not implement spiking neurons, as this would be much harder to train and scale. We sincerely hope that by showing that the dynamical view of neurons improves performance in a wide spectrum of tasks, our paper will motivate following subsequent work not only in deep learning and AI but also in neuroscience and physics.
>
> > ***For table 3 results are presented with error bars over 5 seeds (which I favour), why is this not the case for any of the other results presented?***
>
> We will try to repeat experiments with different random seeds. However, because of our limited computational resources compared to the number of tasks and models we tested in our work, it will take some time to complete and that is why we could compute mean and stds for the Sudoku task, where the training completes much faster than the other tasks.  We will report the error bars of each performance as much as possible until the end of the rebuttal phase.
>
> > ***could you clarify to me the exact novelty of the model? Is it that it's not been introduced in deep neural nets before?***
>
> Thank you for the comment! As far as we know, the original Kuramoto model and its multidimensional model have not been used in deep nets before (this work [ii] uses the original Kuramoto model for clustering but uses a single layer (the connectivity is predicted by a conventional multilayer neural network) in very synthetic cases). The model’s novelty is as follows:
> - The use of the multidimensional Kuramoto model. (Table 7 and Figure 14 in the updated manuscript show that models with N=2, which is close to the original Kuramoto model, significantly underfit in the object discovery experiments and the Sudoku solving.)
> - The building block we propose. It is composed of a pair of the K-layer and the readout module, which together process the conditional stimuli and oscillators. Without norm terms, for example, the model is unstable and again underfits the CIFAR10 classification problem (Figure 16). The same figure shows that just stacking Kuramoto layers without conditional stimuli further underfits the task.
> - The overall network architecture with the proposed blocks. We solve a wide variety of tasks with a shared architecture and show improvement over transformers and conventional ConvNets.
>
> Our work’s novelty is not limited to the model architecture. We demonstrate that the binding and neural synchrony-based idea, which have been shown effective only on relatively synthetic tasks, can be extended to natural images in object discovery tasks with competitive performance to the well-used slot-attention, and that also it proves to be effective in very different tasks such as reasoning and robustness.

---

> ### Author Response · Authors · 2024-11-21
> **Response to reviewer ynsZ**
>
> > ***line 308 'our..models differ from these approaches in various aspects such as...' should really provide all fundemental differences beyond some examples. In fact, in any case, what's the difference between 'asymmetric connections' and 'their symmetry breaking term'?***
>
> Asymmetric connection means J is an asymmetric matrix. Usually, as in the standard Ising model and Kuramoto model, $J_{ij}$ is symmetric: $J_{ij} = J_{ji}^{\rm T}$. This symmetry is important to guarantee the model’s update to minimize a certain energy function. For example, Energy transformer[iii] introduces a symmetrized attention mechanism to have such a guarantee. However, we find such symmetric models basically worsen the performance compared to asymmetric ones (See the response to reviewer AtLx https://openreview.net/forum?id=nwDRD4AMoN&noteId=dPzqDmDXMp). The symmetry-breaking term is $C$, which is a data-dependent term and not introduced in the "Hopfield networks is all you need"[iv] and [iii] and found to be important to stably train AKOrN on the CIFAR10 classification task (Figure 16).
> Thank you for the suggestion! We extend the discussion of [iii] and [iv] in the updated manuscript.
>
> >***line 123 (and other places): what's C? the size of the neural population? this should be clarified***
>
> Sorry for the confusion. Yes, it’s the number of oscillators. We added the description of what $C$ is in the latest manuscript.
>
> > ***line 245: what is 'proper' energy?***
>
> The energy is proper if the differential equation is guaranteed to minimize that energy. In our case, where we allow asymmetric connection in $J$, Eq(2) is not guaranteed to minimize the energy defined in Eq(3). However, it is empirically shown that Eq(2) moves the oscillators roughly in the decreasing direction of Eq(3) (Please see animations in Supplementary Material, which includes plots of energy values over timesteps demonstrating this behavior).
>
> > ***line 323: I don't understand how these convolution/self-attention layers are iterated upon. What are the shared parameters?***
>
> The parameters are shared across Kuramoto steps within the same layer but differ across different layers. For example, there are three layers in the network trained on ImageNet used in the object discovery experiment, and each layer has its own learnable parameters for both the Kuramoto layers and the Readout layers.
> We noticed that we had forgotten to put the layer index on $J$ and $\Omega$ in Eq(4). We fixed it in the updated manuscript. Thank you for raising the question!
>
> > ***I didn't understand the implementation for the sudoku task. 1. is the entire 9x9 soduku grid fed all at once, or one digit at a time?***
>
> Sorry for the confusion. The former is what we did. We see each Sudoku grid as a 9x9 resolution image and process it similarly to the AKOrNs used in the object discovery experiments.
>
> > ***how come x_i takes the value c_i when a digit is given, does the model no longer follow equation 2? Or is this just the initial value taken before the Kuramoto steps?***
>
> We apologize for the lack of explanation about this. $x_i$ is initialized as “a normalized $c_i$”: $c_i/||c_i||_2 $ if a digit is given, since $c_i$ itself is not constrained on the sphere. Otherwise, $x_i$ is initialized randomly from the uniform distribution on the sphere. This initialization is done only once to create the initial $X$ fed into the network, and every $x_i$ is updated following Eq. (4) and (5) after the initialization. We fixed the sentence describing this initialization in the updated manuscript.
>
> > ***line 425: 'the energy is a good indication of the solution's correctness'. Why? I can believe that stability is often a desirable thing in tasks but think this line deserves some elaboration***
>
> Please kindly refer to the General comments, where we elaborate on how the energy can indicate the correctness of the solution.
>
> >***would it be possible to give a brief description/overview of what 'common corruptions' are?***
>
> Thank you for your suggestions! We include an example image of each corruption type in Figure 21 in the Appendix. The images are artificially generated corruptions that mimic naturally occurring perturbations, such as blur, weather, digital noises, and so on.  Please refer to this work [v] for details.
>
> > ***why is it surprising that confidence and accuracy are linearly correlated?***
>
> Thank you for the comment! We rephrased it with almost perfect correlation. Usually, neural network models without regularization are overconfident in their predictions, as we show the ResNet's confidence vs. acc plot at the left bottom in Figure 9. On the other hand, regularized models can be the opposite and be too conservative, like the models on the top rows in the same figure. Our model achieves very high uncertainty quantification without any explicit regularization, which is rarely seen.

---

> > ### Author Response · Authors · 2024-11-21
> > **Response to reviewer ynsZ**
> >
> > > The format of figure references could be improved. I would recommend references as 'Fig. X'. Currently there is either the period missing or there's no space (e.g. 'Fig1' on line 96). Same with Tab X -> Table X. Should also be Figs for multiple figures
> >
> > Thank you for pointing that out. We change the references to Fig. and Tab., and to Figs and Tabs for multiple figures and Tables in the update manuscript.
> >
> > > the notation x_{chw} is confusing, I think x_{c,h,w} is better
> >
> > Thank you for your suggestions, we fixed our notation accordingly.
> >
> > > on line 201 the terms should be switched around (the initial state of the next block is the thing being defined). But also, how does this fit in with line 189 where X^(l,0) = X^l?
> >
> > Thanks for pointing that out. This is a typo and $X^{(l,0)} = X^{(l-1)}$ is correct. We fixed it.
> >
> > > the metrics in all of the tables should be clarified. I presume the unit is %, but e.g. in the case of Table 1 I'm not entirely sure what the metric is. % objects correctly identified?
> >
> > Thank you for your suggestion! Let us clarify each metric here:
> > - The Adjusted Rand Index (ARI) computes how well the clusters align with object masks compared to random cluster assignments. The foreground ARI (FG-ARI) only considers foreground objects and is a well-used metric in object discovery tasks. The maximum value of 100 indicates perfect alignment between the obtained clusters and the object masks. If the cluster assignment is completely random or all features are assigned to the same cluster, the value is 0. (See https://scikit-learn.org/dev/modules/clustering.html#adjusted-rand-score for details.)
> > - The Mean-Best-Overlap (MBO) first assigns each cluster to the highest overlapping ground truth mask and then computes the average intersection-over-union (IoU) of all pairs. The value takes 100 at maximum. Following the literature, we exclude the background mask from the MBO evaluation. Since MBO computes IoU, tightly aligned object masks result in a higher score compared to FG-ARI (FG-ARI does not penalize the mask that extends into the background region).
> >
> > We put those explanations in Section D.1.1 in the Appendix and refer to them in the main texts in the updated manuscript.
> >
> > > in the results section the language can be overly strong. For example 'far better' (line 377) and 'vastly better' (line 416) do not appear to me as very scientific language.
> >
> > Thank you for your suggestion. We change them to moderate ones (far better -> considerably better”, “vastly better” -> “better”).
> >
> > > in the discussion there is talk of small N resulting in a strongly regularized model. Given that the size of N nor regularization was mentioned in the prior text, this comes rather abruptly as is confusing to the reader
> >
> > Sorry for the confusion and thank you for the suggestion. We decided to incorporate that paragraph into the experimental setting section in Section D in the Appendix.
> >
> > > line 150: 'set C a silhouette' -> 'set C as a silhouette'
> >
> > Thank you for pointing it out. We fixed it.
> >
> >
> > ## References
> > - [i] Bard E. and Kleinfeld D. "Traveling electrical waves in cortex: insights from phase dynamics and speculation on a computational role." Neuron 29.1 (2001): 33-44.
> > - [ii] Ricci et al. Kuranet: systems of coupled oscillators that learn to synchronize. arXiv preprint arXiv:2105.02838, 2021
> > - [iii] Hoover et al. Energy transformer. In Advances in Neural Information Processing Systems (NeurIPS), 2023.
> > - [iv] Ramsauer et al. Hopfield networks is all you need. arXiv.org, 2020.
> > - [v] Hendrycks D. and  Dietterich T. Benchmarking neural network robustness to common corruptions and perturbations. In Proc. of the International Conf. on Learning Representations (ICLR), 2019.

---

> > > ### Comment · Reviewer_ynsZ · 2024-11-26
> > >
> > > Thank you authors for your response.
> > >
> > > Overall I have a clearer intepreation of the paper. Now that the novelty of the model is clear to me (Kuramoto neurons have not to the authors' knowledge been used in neural nets) I am more confident in the positive impact this paper can have. I also do agree with the authors that it is fair to only compare with models without large, pre-trained image networks, against which the model's capabilities are notable.
> > >
> > > I am also affected by other reviewers' enthusiasm for this work, and see no reason why it should not be accepted. I am duly increasing my score.

---

> ### Author Response · Authors · 2024-11-27
>
> Thank you for your comments and feedback, and for substantially raising the score!  We are delighted that our response has clarified your questions.
>
> Regarding the error bars; we conducted object discovery experiments with different random seeds for the weight initialization and show the mean and std of the object discovery performance metrics in Figure 3 and Table 1 in the main sections as well as Tables 14-17 in the appendix in the latest manuscript. We see that any updated values do not affect our claims.
>
> Because we have less than one day before the revision deadline, we will not be able to add error bars for the object discovery experiments on natural images or the image classification task on CIFAR-10. However, for those tasks, we believe that models initialized with different random seeds will not exhibit significant performance differences that could impact our claims. This is because the model for the natural image object discovery is trained on ImageNet, which contains a relatively larger number of images compared to other datasets, leading to smaller variance across differently initialized models. Additionally, the distinction between AKOrNs and conventional models in the image classification task is significantly evident in performance metrics such as adversarial robustness and uncertainty calibration.
>
> We thank you again for your suggestions and feedback, which greatly improve the quality and clarity of our manuscript!

---

### Official Review · Reviewer_uqq5 · 2024-11-04

**Soundness:** 3
**Presentation:** 3
**Contribution:** 3
**Rating:** 8
**Confidence:** 4

**Summary:**

In this work, the authors proposed to use Kuramoto oscillatory neurons (AKOrN) to replace the usual thresholding units in deep neural networks. The basic neuron architecture can be applied to layers with different connectivity patterns, including convolutional and self-attention layers. Models with Kuramoto oscillatory neurons show promising performance in visual discovery and reasoning tasks, they are also more robust to adversarial attacks and input corruptions.

**Strengths:**

1. The neuronal architecture draws inspiration from physics and neuroscience and is very novel in deep learning.
2. Model performance is impressive across a range of tasks.
3. Extensive experiments show that the method applies to mainstream architectures including convolutional neural networks and transformers.

**Weaknesses:**

1. While AKOrN shows promising performance on the selected tasks, the model might not work as well on more "classical tasks" such as image classification.
2. Although the motivation of AKOrN is clear from a neuroscience perspective, how, and why the model shows superior performance in the tested task is not well understood.

**Questions:**

1. Judging from the method, AKOrN seems to be much more computationally costly than the usual thresholding units. I noticed that the authors controlled the number of channels in AKOrN models, but showing the training/inference time would be helpful.
2. It would be helpful to include some explicit discussion on the weakness of the method. i.e., what tasks are probably more suitable for this method and why?
3. As mentioned above, more analysis might help establish the advantage of AKOrN. For example, the paper shows that the AKOrN models show better robustness, but it is not clear from the design that they are more robust. Thus analysis of the network dynamics on some toy tasks might be helpful (e.g., synthesized classification tasks on low-dim vectors).

---

> ### Author Response · Authors · 2024-11-21
> **Response to reviewer uqq5**
>
> Thank you very much for the review and suggestions! We address your concerns one by one.
>
> - ***Judging from the method, AKOrN seems to be much more computationally costly than the usual thresholding units. ... showing the training/inference time would be helpful.***
>
> Thank you for your suggestion, and please see the General comments regarding this matter. In summary, the runtime overhead in AKOrN can be negligible as the model size increases. The runtime comparison including both training and inference is shown in Figure 17.
>
> - ***the paper shows that the AKOrN models show better robustness, but it is not clear from the design that they are more robust. Thus analysis of the network dynamics on some toy tasks might be helpful.***
>
> We conducted an analysis of the oscillator states of networks trained on CIFAR10 and found that oscillators highly fluctuate over timesteps, which we could think prevents the model from learning noise-sensitive features. Please see the General comments for details.
>
> Finally, seemingly these three questions and concerns are closely related together, so we answer in the following altogether.
> - ***the model might not work as well on more "classical tasks" such as image classification.***
> - ***why the model shows superior performance in the tested task is not well understood.***
> - ***what tasks are probably more suitable for this method and why?***
>
> While several aspects of the models are not fully uncovered (such as adversarial robustness), the efficacy of the object discovery and combinatorial problems is intuitive. The Kuramoto model promotes synchronization in oscillatory neurons, where synchronized neurons form a cluster. These clusters are then processed by subsequent Kuramoto updates that further generate more abstract features. This idea is a motivation to use the model to learn binding object features, as each object can be seen as elements that are clustered, mimicking the visual recognition process where objects are grouped based on shared features. The use of the Kuramoto model for reasoning tasks is also reasonable as the Kuramoto model can be seen as a continuous Ising model. The Ising model has been used to solve combinatorial problems.
> To address your questions regarding suitable tasks, we believe our model is well-suited for solving _complex visual reasoning tasks_. This is because the ability to learn abstract (object) features and reason over those features is crucial for such tasks. Studying our model's efficacy in such tasks is left for future work.

---

> > ### Comment · Reviewer_uqq5 · 2024-11-26
> >
> > Thanks for addressing my concerns. I believe this would be a valuable contribution to the conference I am more confident in my assessment now. I am also looking forward to follow-up works that study the efficacy of the neuronal architecture in a wider range of tasks.

---

### Author Response · Authors · 2024-11-21
**General comments**

We owe great thanks to all reviewers for their helpful comments and suggestions for improving our manuscripts!
We have revised our manuscript since the initial submission, incorporating reviewers’ suggestions and including updated results for the object discovery tasks.

Let us highlight the changes we made:
1. Added **Model analysis** section in the Appendix, where we provide an extensive comparison of various model designs. This section addresses some of the concerns and questions raised by the reviewers.
2. Improved the object discovery performance by **up-tiling**. Up-tiling enables us to compute finer object assignments on SSL features, and substantially improves SSL models’ performance including our AKOrN. AKOrN is now better than any other prior works on PascalVOC, and is better on COCO than existing methods except for a recent model SPOT (See Figures 4,5 and Tables 1,2 for the improved results and Section D.1.2 for the methodological details of up-tiling).

---

We here address some concerns that were commonly raised by different reviewers.

- ***Regarding the computational overhead in AKOrN*** (by reviewers DAWQ and uqq5)

The main computational overheads in AKOrN are the projection and normalization in Eq.(4) and Eq.(5). The computational order of those operations is linear w.r.t the feature sizes, thus the overhead becomes ignorable as the model size increases. The below table shows the training and inference time of AKOrNs and their non-Kuramoto counterpart models in different tasks. Training time is the time taken to complete a single gradient step (excluding data loading). Inference time is the time taken for a single forward pass with a mini-batch size of 100.  The unit is milliseconds.
| Dataset   | ItrConv | AKOrN |
|----------|-----------------:|-----------------:|
| CIFAR10 (Train)  | 30.72         | 52.84             |
| CIFAR10 (Infer)   | 8.84          | 13.58           |

| Dataset   | ItrSA   | AKOrN  |
|----------|-----------------:|-----------------:|
|   Sudoku (Train) | 88.18           | 96.59              |
|   Sudoku (Infer)  |  29.00       | 33.40           |


| Dataset  | ItrSA  | AKOrN |
|----------|------------------:|------------------:|
|   CLEVRTex (Train) | 637.10           | 661.50             |
|  CLEVRTex (Infer)  |   41.40        | 45.50            |

 In CIFAR-10 experiments using relatively small convolutional networks, AKOrN requires approximately 1.7 times the training computation time compared to its non-Kuramoto counterparts. However, in experiments with larger channel sizes, such as the CLEVRTex object discovery and the Sudoku solving, the overhead drops significantly to 1.1x and 1.04x, respectively. For inference, the additional computational cost with AKOrN is slightly higher compared to training but remains modest, at 1.15x for CLEVRTex and 1.09x for Sudoku. We include these runtime comparisons in the updated manuscript as Figure 17.

- ***Regarding the rationale of the robustness by AKOrN*** (by reviewers uqq5, AtLx, and ynsZ)

One possible explanation for AKOrN’s robustness is the fluctuations occurring in the oscillatory states.  We observe that, especially in the first layer, each Kuramoto update generates _noise-like_ perturbations (Please see the animation file **cifar10_block1.gif** in the Supplementary Material). This _deterministic noise_ would help the model avoid learning features vulnerable to perturbations. Interestingly, simply adding Gaussian noise during ResNet training does not produce models as robust as AKOrN. The table below compares the robustness of ResNets trained with Gaussian noise to that of AKOrN. We tuned the noise strength to match the clean test set accuracy of AKOrN and selected two ResNet models with similar clean accuracy. Across all robustness metrics, AKOrN consistently outperforms the ResNets, particularly excelling in adversarial robustness. The same table is shown in Table 8, and a visualization of the fluctuations is provided in Figure 15.

| Model              | Clean Accuracy ↑ | Adv Accuracy ↑ | CC Accuracy ↑ | ECE CC ↓ |
|--------------------|------------------|----------------|---------------|----------|
| ResNet (σ = 0.2)   | 85.2            | 22.3           | 75.1          | 2.3      |
| ResNet (σ = 0.225) | 83.9            | 25.5           | 73.6          | 2.6      |
| AKOrN (N = 2)      | 84.6            | **64.9**       | **78.3**      | **1.8**  |

---

> ### Author Response · Authors · 2024-11-21
> **Official comments (cont.)**
>
> - ***Why the energy value indicates the solution's correctness?*** (by reviewers AtLx and ynsZ)
>
> (Note: $x_i, y_i, c_i$ and $J_{ij}$ are multidimensional vectors and a connectivity matrix, respectively, even if they are written in non-bold style. It seems that ICLR's markdown cannot process bold symbols for some reason)
>
> Although the update is not guaranteed to minimize the energy, we still have a good interpretation of what the value in Eq(3) means. Please see the Figure 11. in the updated manuscript. The figure shows that the negative energy  $ \langle c_i + \sum_j  J_{ij} x_j, x_i \rangle$ and the length of the projected update are ***inversely proportional*** given the length of $y_i:=c_i + \sum_{j}  J_{ij} x_j$.
> More specifically, the eq(2) can be rewritten as
>
> $$ E(X) = - \sum_i c_i^{\rm T} x_i - \sum_{ij} x_i^{\rm T} J_{ij} x_j  = - \sum_i  ( c_i + \sum_j J_{ij} x_j)^{\rm T} x_i  = -\sum_i  \langle y_i, x_i \rangle $$
>
> Thus, the negative energy tells us how the update direction is aligned with the current oscillators' state. If it’s small, the length of the projected update in the oscillators is expected to be high, and vice versa. From this interpretation, we see the energy as a sensible indicator of how the oscillators settle down, especially for the Sudoku problem where the solution is well-defined. It is natural to consider that low-energy states represent the model’s confidence in the solutions, as these oscillator states are less likely to undergo further updates. However, if the model fails to learn a good energy landscape, then even low-energy states may correspond to incorrect solutions. This _incorrectly learned energy_ happens when we do not use the natural frequency term $\Omega$ (See Figure 26 and Table 18 in the Appendix). The natural frequency term, which is a completely asymmetric transformation, serves as a form of exploration in the energy landscape, which would allow the states to escape from bad minima.

---

### Meta-Review · Area_Chair_mS6V · 2024-12-10

**Metareview:**

This paper takes on a long-standing idea from the computational neuroscience community, namely, that binding of different features in an input can be done using oscillations in unit activity that synchronize to indicate binding. The authors propose Artificial Kuramoto Oscillatory Neurons (AKOrN), which employs oscillatory units based on the Kuramoto (1984) model.

The authors claim that their system is capable of binding together features in an input (e.g. objects in an image). They further claim that this provides performance improvements across various tasks such as unsupervised object discovery, adversarial robustness, calibrated uncertainty quantification, and reasoning. They demonstrate the vailidity of these claims using a range of benchmarks, such as CLEVRTex, PascalVOC, and Sodoku puzzles.

The primary strength of this paper is that it manages to solve a long-standing technical issue (how to bind features) with a brain-inspired solution that no one else has previously been able to get working really well. It also provides strong empirical support for the claims, and illuminating analyses of the model. There were some weaknesses with clarity on the model's computational overhead and critical parts, but those were addressed in rebuttal. Altogether, it is a very strong paper. As such, a decision to accept for an oral was reached.

**Additional Comments On Reviewer Discussion:**

The reviews for this paper were quite positive. The reviewers raised some concerns, including the following notable ones:

- Computational overhead of the method
- Importance of different components of the model design
- Comparison to existing approaches
- Explanation for robustness results

The authors did a good job of addressing the concerns and three of the four reviewers replied to the rebuttal with recognition of their work. The one reviewer that didn't respond (DAWQ) had already given the paper a 10, so their response was arguably not necessary.

In the end, based on the high initial and final scores, and enthusiasm of the reviewers, a decision to accept for an oral was natural.

---

### Decision · Program_Chairs · 2025-01-22

Accept (Oral)